# STOCHASTIC LATENT RESIDUAL VIDEO PREDICTION

## ABSTRACT

Video prediction is a challenging task: models have to account for the inherent uncertainty of the future. Most works in the literature are based on stochastic image-autoregressive recurrent networks, raising several performance and applicability issues. An alternative is to use fully latent temporal models which untie frame synthesis and dynamics. However, no such model for video prediction has been proposed in the literature yet, due to design and training difficulties. In this paper, we overcome these difficulties by introducing a novel stochastic temporal model. It is based on residual updates of a latent state, motivated by discretization schemes of differential equations. This first-order principle naturally models video dynamics as it allows our simpler, lightweight, interpretable, latent model to outperform prior state-of-the-art methods on challenging datasets.

## 1 INTRODUCTION

Being able to predict the future of a video from a few conditioning frames in a self-supervised manner has many applications in fields such as reinforcement learning (Gregor et al., 2019) or robotics (Babaeizadeh et al., 2018). More generally, it challenges the ability of a model to capture visual and dynamic representations of the world. Video prediction has received a lot of attention from the computer vision community. However, most proposed methods are deterministic, reducing their ability to capture video dynamics, which are intrinsically stochastic (Denton & Fergus, 2018).

Stochastic video prediction is a challenging task which has been tackled by recent works. Most state-of-the-art approaches are based on image-autoregressive models (Denton & Fergus, 2018; Babaeizadeh et al., 2018), built around Recurrent Neural Networks (RNNs), where each generated frame is fed back to the model to produce the next frame. However, performances of their temporal models innately depend on the capacity of their encoder and decoder, as each generated frame has to be re-encoded in a latent space. Such autoregressive processes induce a high computational cost, and strongly tie the frame synthesis and temporal models, which may hurt the performance of the generation process and limit its applicability (Gregor et al., 2019; Rubanova et al., 2019).

An alternative approach consists in separating the dynamic of the state representations from the generated frames, which are independently decoded from the latent space. In addition to removing the aforementioned link between frame synthesis and temporal dynamics, this is *computationally appealing* when coupled with a low-dimensional latent-space. Moreover, such models can be used to *shape a complete representation* of the state of a system, e.g. for reinforcement learning applications (Gregor et al., 2019), and more interpretable than autoregressive models (Rubanova et al., 2019). Yet, these State-Space Models (SSMs) are more difficult to train as they require non-trivial latent state inference schemes (Krishnan et al., 2017) and a careful design of the dynamic model (Karl et al., 2017). This leads most successful SSMs to only be evaluated on small or artificial toy tasks.

In this work, we propose a *novel stochastic dynamic model* for the task of video prediction which successfully leverages structural and computational advantages of SSMs that operate on low-dimensional latent spaces. The dynamic component determines the evolution through *residual updates* of the latent state, conditioned on learned stochastic variables. This formulation allows us to implement an efficient training strategy and process in an interpretable manner complex high-dimensional data such as videos. This residual principle can be linked to recent advances relating residual networks and Ordinary Differential Equations (ODEs) (Chen et al., 2018). This interpretation opens new perspectives such as generating videos at different frame rates, as demonstrated in our experiments. Overall, this approach outperforms current state-of-the-art models on the task of stochastic video prediction, as demonstrated by comparisons with competitive baselines on representative benchmarks.

## 2    RELATED WORK

Video synthesis covers a range of different tasks, such as video-to-video translation (Wang et al., 2018), super-resolution (Caballero et al., 2017), interpolation between frames (Jiang et al., 2018), unconditonal generation (Tulyakov et al., 2018), or video prediction, which is the focus of this paper.

**Deterministic models.**    Inspired by prior sequence generation models using RNNs (Graves, 2013), a number of video prediction methods (Srivastava et al., 2015; Villegas et al., 2017; Wichers et al., 2018) rely on LSTMs (Hochreiter & Schmidhuber, 1997), or, like Ranzato et al. (2014) and Jia et al. (2016), on derived networks such as ConvLSTMs (Shi et al., 2015) taking advantage of Convolutional Neural Networks (CNNs). Indeed, computer vision approaches are usually tailored to high-dimensional video sequences and propose domain-specific techniques as they often use pixel-level transformations and optical flow (Shi et al., 2015; Walker et al., 2015; Finn et al., 2016; Jia et al., 2016; Vondrick & Torralba, 2017; Liang et al., 2017; Liu et al., 2017; Lotter et al., 2017; Lu et al., 2017a; Fan et al., 2019) that help to produce high-quality predictions. Such predictions are, however, deterministic, thus hurting their performance as they fail to generate sharp long-term video frames (Babaeizadeh et al., 2018; Denton & Fergus, 2018). Following Mathieu et al. (2016), some works proposed to use an adversarial loss (Goodfellow et al., 2014) on the predictions of their model to sharpen the generated frames (Vondrick & Torralba, 2017; Liang et al., 2017; Lu et al., 2017a; Xu et al., 2018). Nonetheless, adversarial losses are notoriously hard to train, and lead to mode collapse, preventing diversity of generations.

**Stochastic and image-autoregressive models.**    Some approaches rely on exact likelihood maximization, using pixel-level autoregressive generation (van den Oord et al., 2016; Kalchbrenner et al., 2017) or normalizing flows through invertible transformations between the observation space and a latent space (Kingma & Dhariwal, 2018; Kumar et al., 2019). However, they require careful design of complex temporal generation schemes manipulating high-dimensional data, thus inducing a prohibitive temporal generation cost. More efficient continuous models rely on Variational Auto-Encoders (VAEs) (Kingma & Welling, 2014; Rezende et al., 2014) for the inference of low-dimensional latent state variables. Except Xue et al. (2016) who learn a one-frame-ahead VAE, they model sequence stochasticity by incorporating a random latent variable per frame into a deterministic RNN-based image-autoregressive model. Babaeizadeh et al. (2018) integrate stochastic variables into the ConvLSTM architecture of Finn et al. (2016). Concurrently with He et al. (2018), Denton & Fergus (2018), with Castrejon et al. (2019) in a follow-up, use a prior LSTM conditioned on previously generated frames in order to sample random variables that are fed to a predictor LSTM. Finally, Lee et al. (2018) combine the ConvLSTM architecture and this learned prior, adding an adversarial loss on the predicted videos to sharpen them at the cost of a diversity drop. Yet, all these methods are image-autoregressive, as they feed their predictions back into the latent space, thus tying the frame synthesis and temporal models and increasing their computational cost. Concurrently to our work, Minderer et al. (2019) propose to use the autoregressive VRNN model (Chung et al., 2015) on learned image key-points instead of raw frames. While this change could mitigate the aforementioned problems, the extent of such mitigation is unclear. We follow a complementary approach by directly proposing a dynamic model that is state-space and acts on a small latent state, tackling these issues.

**State-space models.**    Many latent state-space models have been proposed for sequence modelization (Bayer & Osendorfer, 2014; Fraccaro et al., 2016; 2017; Krishnan et al., 2017; Karl et al., 2017; Hafner et al., 2019), usually trained by deep Variational Inference (VI). These methods, which use locally linear temporal transition functions or RNN-based dynamics, are designed for and tested on low-dimensional data, as learning such models on complex data is challenging, or focus on control or planning tasks. In contrast, our fully latent method is the first one to be successfully applied to complex high-dimensional data such as videos, thanks to a temporal model based on residual updates of its latent state. It thus falls within the scope of a recent trend linking differential equations with neural networks (Lu et al., 2017b; Long et al., 2018), leading to the integration of ODEs, that are seen as continuous residual networks, in neural network architectures (Chen et al., 2018). However, the latter work and follow-ups (Rubanova et al., 2019; Yıldız et al., 2019) are either limited to low-dimensional data, prone to overfitting or unable to handle stochasticity within a sequence. Another line of works considers stochastic differential equations (SDEs) with neural networks (Ryder

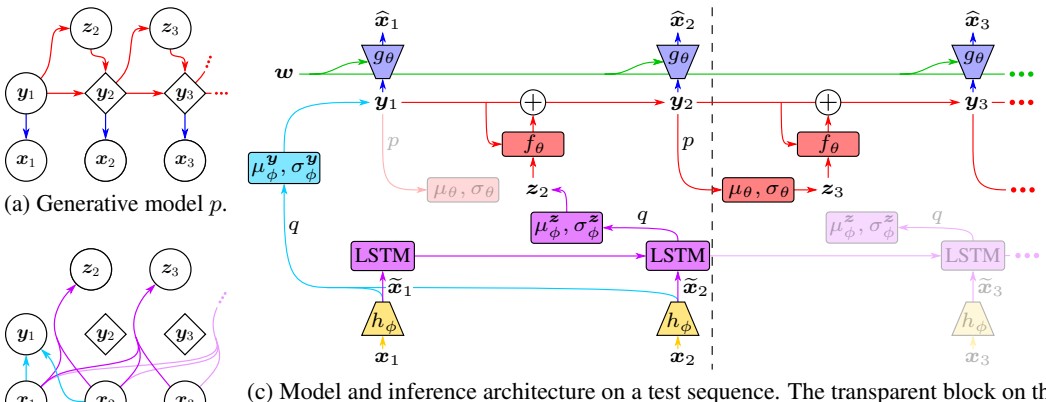

(a) Generative model $p$.

(b) Inference model $q$.

(c) Model and inference architecture on a test sequence. The transparent block on the left depicts the prior, and those on the right correspond to the full inference performed at training time.

Figure 1: (a), (b) Proposed generative and inference models. Diamonds and circles represent, respectively, deterministic and stochastic states. (c) Corresponding architecture with two parts: inference on conditioning frames on the left, generation for extrapolation on the right. $h_\theta$ and $g_\theta$ are deep CNNs, and other named networks are Multilayer Perceptrons (MLPs).

et al., 2018; De Brouwer et al., 2019), but are limited to continuous Brownian noise, whereas video prediction additionally requires to model punctual stochastic events.

## 3 MODEL

We consider the task of stochastic video prediction, consisting in approaching, given a number of conditioning video frames, the distribution of possible future frames given this conditioning.

### 3.1 LATENT RESIDUAL DYNAMIC MODEL

Let $\boldsymbol{x}_{1:T}$ be a sequence of $T$ video frames. We model their evolution by introducing latent variables $\boldsymbol{y}$ that are driven by a dynamic temporal model. Each frame $\boldsymbol{x}_t$ is then generated from the corresponding latent state $\boldsymbol{y}_t$ only, making the dynamics independent from the previously generated frames.

We propose to model the transition function of the latent dynamic of $\boldsymbol{y}$ with a *stochastic residual network*. State $\boldsymbol{y}_{t+1}$ is chosen to deterministically depend on the previous state $\boldsymbol{y}_t$, conditionally to an auxiliary random variable $\boldsymbol{z}_{t+1}$. These auxiliary variables encapsulate the randomness of the video dynamics. They have a learned factorized Gaussian prior that depends on the previous state only. The model is depicted in Figure 1a, and defined as follows:

$$\boldsymbol{y}_1 \sim \mathcal{N}(\boldsymbol{0}, I), \ \boldsymbol{z}_{t+1} \sim \mathcal{N}\big(\mu_\theta(\boldsymbol{y}_t), \sigma_\theta(\boldsymbol{y}_t)I\big), \ \boldsymbol{y}_{t+1} = \boldsymbol{y}_t + f_\theta(\boldsymbol{y}_t, \boldsymbol{z}_{t+1}), \ \boldsymbol{x}_t \sim \mathcal{G}\big(g_\theta(\boldsymbol{y}_t)\big), \quad (1)$$

where $\mu_\theta$, $\sigma_\theta$, $f_\theta$ and $g_\theta$ are neural networks, and $\mathcal{G}\big(g_\theta(\boldsymbol{y}_t)\big)$ is a probability distribution parameterized by $g_\theta(\boldsymbol{y}_t)$. In our experiments, $\mathcal{G}$ is a normal distribution with fixed diagonal variance and mean $g_\theta(\boldsymbol{y}_t)$. Note that $\boldsymbol{y}_1$ is assumed to have a standard Gaussian prior, and, in our VAE setting, will be inferred from conditioning frames for the prediction task, as shown in Section 3.3.

The residual update rule takes inspiration in the Euler discretization scheme of differential equations. The state of the system $\boldsymbol{y}_t$ is updated by its first-order movement, i.e., the residual $f_\theta(\boldsymbol{y}_t, \boldsymbol{z}_{t+1})$. Compared to a regular RNN, this simple principle makes our temporal model lighter and more interpretable. Equation (1), however, differs from a discretized ODE because of the introduction of the stochastic discrete-time variables $\boldsymbol{z}$. Nonetheless, we propose to allow the Euler step size $\Delta t$ to be smaller than 1, as a way to make the temporal model closer to a continuous dynamics. The updated dynamics becomes, with $\frac{1}{\Delta t} \in \mathbb{N}$ to synchronize the step size with the video frame rate:

$$\boldsymbol{y}_{t+\Delta t} = \boldsymbol{y}_t + \Delta t \cdot f_\theta\big(\boldsymbol{y}_t, \boldsymbol{z}_{\lfloor t \rfloor + 1}\big). \quad (2)$$

For this formulation, the auxiliary variable $\boldsymbol{z}_t$ is kept constant between two integer time steps. Note that a different $\Delta t$ can be used during training or testing. This allows our model to generate videos at

an arbitrary frame rate since each intermediate latent state can be decoded in the observation space. This ability enables us to observe the quality of the learned dynamic as well as challenge its ODE inspiration by testing its generalization to the continuous limit in Section 4. In the following, we consider $\Delta t$ as a hyperparameter. For the sake of clarity, we consider that $\Delta t = 1$ in the following; generalizing to smaller $\Delta t$ is straightforward as Figure 1a remains unchanged.

## 3.2 CONTENT VARIABLE

Some components of video sequences can be static, such as the background or shapes of moving objects. They may not impact the dynamics; we therefore model them separately, in the same spirit as Denton & Birodkar (2017) and Yingzhen & Mandt (2018). We compute a *content variable* $\boldsymbol{w}$ that remains constant throughout the whole generation process and is fed together with $\boldsymbol{y}_t$ into the frame generator. It enables the dynamical part of the model to focus only on movement, hence being lighter and more stable. Moreover, it allows us to leverage architectural advances in neural networks, such as skip connections (Ronneberger et al., 2015), to produce more realistic frames.

This content variable is a deterministic function $c_\psi$ of a fixed number $k < T$ of frames $\boldsymbol{x}_{\mathrm{c}}^{(k)}$:

$$\boldsymbol{x}_{\mathrm{c}}^{(k)} = \boldsymbol{x}_{i_1}, \ldots, \boldsymbol{x}_{i_k}, \qquad \boldsymbol{w} = c_\psi\big(\boldsymbol{x}_{\mathrm{c}}^{(k)}\big) = c_\psi\big(\boldsymbol{x}_{i_1}, \ldots, \boldsymbol{x}_{i_k}\big), \qquad \boldsymbol{x}_t \sim \mathcal{G}\big(g_\theta(\boldsymbol{y}_t, \boldsymbol{w})\big). \quad (3)$$

During testing, $\boldsymbol{x}_{\mathrm{c}}^{(k)}$ are the last $k$ conditioning frames (usually between 2 and 5).

This content variable is not endowed with any probabilistic prior, contrary to the dynamic variables $\boldsymbol{y}$ and $\boldsymbol{z}$. Hence, the information it contains is not constrained in the loss function (see Section 3.3), but only architecturally. To prevent temporal information from leaking in $\boldsymbol{w}$, we propose to uniformly sample these $k$ frames within $\boldsymbol{x}_{1:T}$ during training. We also design $c_\psi$ as a permutation-invariant function (Zaheer et al., 2017), which is done by using an MLP fed with the sum of individual frame representations, similarly to Santoro et al. (2017).

This absence of prior and its architectural constraint allows $\boldsymbol{w}$ to contain as much non-temporal information as possible, while preventing it from containing dynamic information. On the other hand, due to their strong standard Gaussian priors, $\boldsymbol{y}$ and $\boldsymbol{z}$ are encouraged to discard unnecessary information. Therefore, $\boldsymbol{y}$ and $\boldsymbol{z}$ should only contain temporal information that could not be captured by $\boldsymbol{w}$.

Note that this content variable can be removed from our model, yielding a more classical deep state-space model. An experiment in this setting is presented in Appendix E.

## 3.3 VARIATIONAL INFERENCE AND ARCHITECTURE

Following the generative process depicted in Figure 1a, the conditional joint probability of the full model, given a content variable $\boldsymbol{w}$, can be written as:

$$p(\boldsymbol{x}_{1:T}, \boldsymbol{z}_{2:T}, \boldsymbol{y}_{1:T} \mid \boldsymbol{w}) = p(\boldsymbol{y}_1) \prod_{t=1}^{T-1} p(\boldsymbol{z}_{t+1} \mid \boldsymbol{y}_t) p(\boldsymbol{y}_{t+1} \mid \boldsymbol{y}_t, \boldsymbol{z}_{t+1}) \prod_{t=1}^{T} p(\boldsymbol{x}_t \mid \boldsymbol{y}_t, \boldsymbol{w}), \quad (4)$$

where $p(\boldsymbol{y}_{t+1} \mid \boldsymbol{y}_t, \boldsymbol{z}_{t+1}) = \delta\big(\boldsymbol{y}_t + f_\theta(\boldsymbol{y}_t, \boldsymbol{z}_{t+1}) - \boldsymbol{y}_{t+1}\big)$ and $\delta$ is the Dirac delta function centered on $\boldsymbol{0}$, according to the expression of $\boldsymbol{y}_{t+1}$ in Equation (1). Thus, in order to optimize the likelihood of the observed videos $p(\boldsymbol{x}_{1:T} \mid \boldsymbol{w})$, we need to infer latent variables $\boldsymbol{y}_1$ and $\boldsymbol{z}_{2:T}$. This is done by deep Variational Inference using the inference model parameterized by $\phi$ and shown in Figure 1b, which comes down to consider a variational distribution $q_{Z,Y}$ defined and factorized as follows:

$$q_{Z,Y} \triangleq q(\boldsymbol{z}_{2:T}, \boldsymbol{y}_{1:T} \mid \boldsymbol{x}_{1:T}, \boldsymbol{w}) = q(\boldsymbol{y}_1 \mid \boldsymbol{x}_{1:k}) \prod_{t=2}^{T} q(\boldsymbol{z}_t \mid \boldsymbol{x}_{1:t}) \delta\big(\boldsymbol{y}_{t-1} + f_\theta(\boldsymbol{y}_{t-1}, \boldsymbol{z}_t) - \boldsymbol{y}_t\big). \quad (5)$$

This yields the following evidence lower bound (ELBO), whose full derivation is given in Appendix A:

$$\log p(\boldsymbol{x}_{1:T} \mid \boldsymbol{w}) \geq \mathbb{E}_{(\widetilde{\boldsymbol{z}}_{2:T}, \widetilde{\boldsymbol{y}}_{1:T}) \sim q_{Z,Y}} \sum_{t=1}^{T} \log p(\boldsymbol{x}_t \mid \widetilde{\boldsymbol{y}}_t, \boldsymbol{w}) - D_{\mathrm{KL}}\big(q(\boldsymbol{y}_1 \mid \boldsymbol{x}_{1:k}) \,\big\|\, p(\boldsymbol{y}_1)\big)$$

$$- \mathbb{E}_{(\widetilde{\boldsymbol{z}}_{2:T}, \widetilde{\boldsymbol{y}}_{1:T}) \sim q_{Z,Y}} \sum_{t=2}^{T} D_{\mathrm{KL}}\big(q(\boldsymbol{z}_t \mid \boldsymbol{x}_{1:t}) \,\big\|\, p(\boldsymbol{z}_t \mid \widetilde{\boldsymbol{y}}_{t-1})\big) \triangleq \mathcal{L}(\boldsymbol{x}_{1:T}; \boldsymbol{w}, \theta, \phi). \quad (6)$$

The sum of KL divergence expectations implies to consider the full past sequence of inferred states for each time step, due to the dependence on conditionally deterministic variables $\boldsymbol{y}_{2:T}$. However, optimizing $\mathcal{L}(\boldsymbol{x}_{1:T}; \boldsymbol{w}, \theta, \phi)$ with respect to model parameters $\theta$ and variational parameters $\phi$ can be done efficiently by sampling a single full sequence of states from $q_{Z,Y}$ per example, and computing gradients by backpropagation (Rumelhart et al., 1988) trough all inferred variables, using the reparametrization trick (Kingma & Welling, 2014; Rezende et al., 2014). We classically choose $q(\boldsymbol{y}_1 \mid \boldsymbol{x}_{1:k})$ and $q(\boldsymbol{z}_t \mid \boldsymbol{x}_{1:t})$ to be factorized Gaussian so that all KLDs can be computed analytically.

We include an $\ell_2$ regularization term on residuals $f_\theta$ which stabilizes the temporal dynamics of the residual network, as noted by Behrmann et al. (2019) and Rousseau et al. (2019). Given a set of videos $\mathcal{X}$, the full optimization problem, where $\mathcal{L}$ is defined as in Equation (6), is then given as:

$$\arg\max_{\theta, \phi, \psi} \sum_{\boldsymbol{x} \in \mathcal{X}} \left[ \mathbb{E}_{\boldsymbol{x}_{\mathrm{c}}^{(k)}} \mathcal{L}\left( \boldsymbol{x}_{1:T}; c_\psi\left(\boldsymbol{x}_{\mathrm{c}}^{(k)}\right), \theta, \phi \right) - \lambda \cdot \mathbb{E}_{(\boldsymbol{z}_{2:T}, \boldsymbol{y}_{1:T}) \sim q_{Z,Y}} \sum_{t=2}^{T} \big\| f_\theta(\boldsymbol{y}_{t-1}, \boldsymbol{z}_t) \big\|_2 \right]. \quad (7)$$

Figure 1c depicts the full architecture of our temporal model, corresponding to how the model is applied during testing. The first latent variables are inferred with the conditioning framed and are then predicted with the dynamic model. In contrast, during training, each frame of the input sequence is considered for inference, which is done as follows. Firstly, each frame $\boldsymbol{x}_t$ is independently encoded into a vector-valued representation $\widetilde{\boldsymbol{x}}_t$, with $\widetilde{\boldsymbol{x}}_t = h_\phi(\boldsymbol{x}_t)$. $\boldsymbol{y}_1$ is then inferred using an MLP on the first $k$ encoded frames $\widetilde{\boldsymbol{x}}_{1:k}$. Each $\boldsymbol{z}_t$ is inferred in a feed-forward fashion with an LSTM on the encoded frames. Inferring $\boldsymbol{z}$ this way experimentally performs better than, e.g., inferring them from the whole sequence $\boldsymbol{x}_{1:T}$; we hypothesize that this follows from the fact that this filtering scheme is closer to the prediction setting, where the future is not available.

## 4 EXPERIMENTS

This section exposes the experimental results of our method on three standard stochastic video prediction datasets.[1] We compare our method with state-of-the-art baselines on stochastic video prediction. Furthermore, we qualitatively study the dynamics and latent space learned by our model. Training details are described in Appendix C.

The stochastic nature and novelty of the task of stochastic video prediction make it challenging to evaluate (Lee et al., 2018): since videos and models are stochastic, comparing the ground truth and a predicted video is not adequate. We thus adopt the common approach (Denton & Fergus, 2018; Lee et al., 2018) consisting in, for each test sequence, sampling from the tested model a given number (here, 100) of possible futures and reporting the best performing sample against the true video. We report this discrepancy for three commonly used metrics: Peak Signal-to-Noise Ratio (PSNR, *higher is better*), Structured Similarity (SSIM, *higher is better*), and Learned Perceptual Image Patch Similarity (LPIPS, *lower is better*) (Zhang et al., 2018). PSNR tends to promote blurry predictions, as it is a pixel-level measure derived from the $\ell_2$ distance, but greatly penalizes errors in predicted positions of objects in the scenes. SSIM is a similarity metric between image patches. LPIPS is a learned distance between activations of deep CNNs trained on image classification tasks, and have been shown to better correlate with human judgment on real images. While these three metrics are computed frame-wise, the recently proposed Fréchet Video Distance (FVD, *lower is better*) (Unterthiner et al., 2018) aims at directly comparing the distribution of predicted videos with the ground truth distribution through the representations computed by a deep CNN trained on action

---

[1]Code, video samples, and datasets are available at `https://sites.google.com/view/srvp/`.

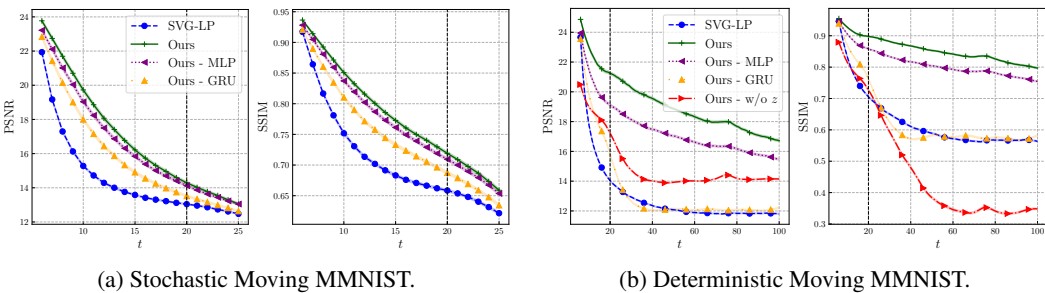

(a) Stochastic Moving MMNIST.      (b) Deterministic Moving MMNIST.

Figure 2: Mean PSNR and SSIM scores with respect to $t$ for all tested models on the SM-MNIST dataset, with their 95%-confidence intervals. Vertical bars mark the length of train sequences.

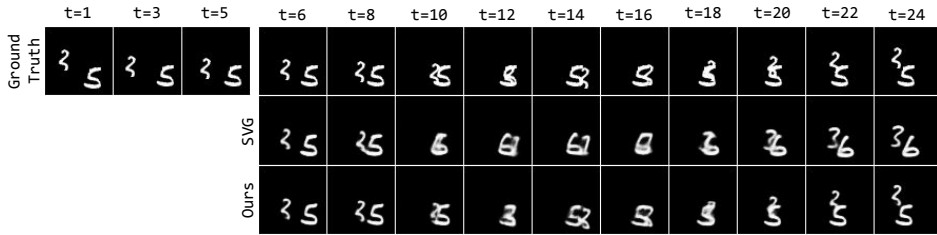

Figure 3: Conditioning frames and corresponding ground truth and best samples with respect to PSNR from SVG and our method for an example of the SM-MNIST dataset.

recognition tasks. It has been shown, independently from LPIPS, to better correlate with human judgment than PSNR and SSIM. We treat all four metrics as complementary, as they capture different modalities. PSNR challenges the dynamics of the predicted videos, while SSIM rather compares local frame patches but loses some dynamics information. LPIPS and FVD both measure the realism of the predictions compared to the ground truth. FVD considers videos as a whole, making it more capable of detecting temporal inconsistencies. On the other hand, the frame-wise LPIPS metric penalizes more the temporal drifts of videos, since it directly compares each predicted and ground truth frame.

We present experimental results on a simulated dataset and two real-world datasets, that we briefly present in the following and detail in Appendix B. The corresponding numerical results can be found in Appendix D. For the sake of concision, we only display a handful of qualitative samples in this section, and refer to Appendix H for additional samples. We compare our model against several state-of-the-art models: SV2P (Babaeizadeh et al., 2018), SVG (Denton & Fergus, 2018) and SAVP (Lee et al., 2018). All baseline results were obtained with pretrained models released by the authors. Note that we use the same neural architecture as SVG for our encoders and decoders in order to perform fair comparisons with this method, which is the closest to ours among the state of the art. Unless specified otherwise, our model is tested with the same $\Delta t$ as in training (see Equation (2)).

**Stochastic Moving MNIST (SM-MNIST).** This dataset consists of one or two MNIST digits (LeCun et al., 1998) moving linearly and randomly bouncing on walls with new direction and velocity sampled randomly at each bounce (Denton & Fergus, 2018). As SV2P and SAVP were not tested on this dataset (in particular, with no pretrain model, code or hyperparameters), we only report scores for SVG as state-of-the-art model on SM-MNIST.

Figure 2a shows quantitative results with two digits. Our model outperforms SVG on both PSNR and SSIM; LPIPS and FVD are not reported as they are not relevant for this synthetic task. Decoupling dynamics from image synthesis allows our method to maintain temporal consistency despite high-uncertainty frames where crossing digits become indistinguishable. For instance in Figure 3, the digits shape changes after they cross in the SVG prediction, while our model predicts the correct digits. To evaluate the predictive ability on a longer horizon, we perform experiments on the classic deterministic version of the dataset (Srivastava et al., 2015). We show the results up to $t + 95$ in

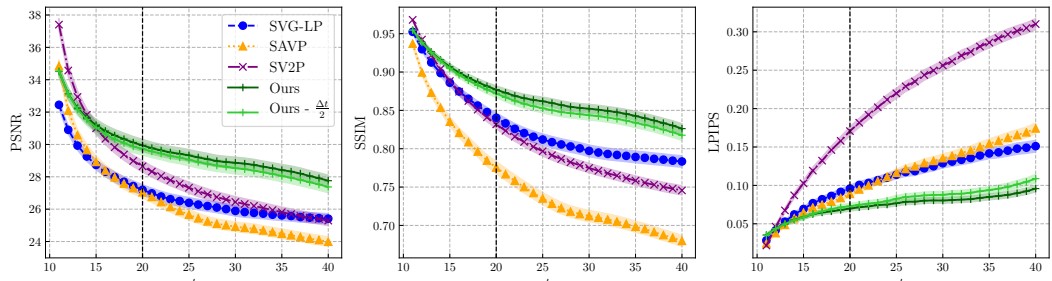

Figure 4: PSNR, SSIM and LPIPS scores with respect to $t$ for all tested models on the KTH dataset.

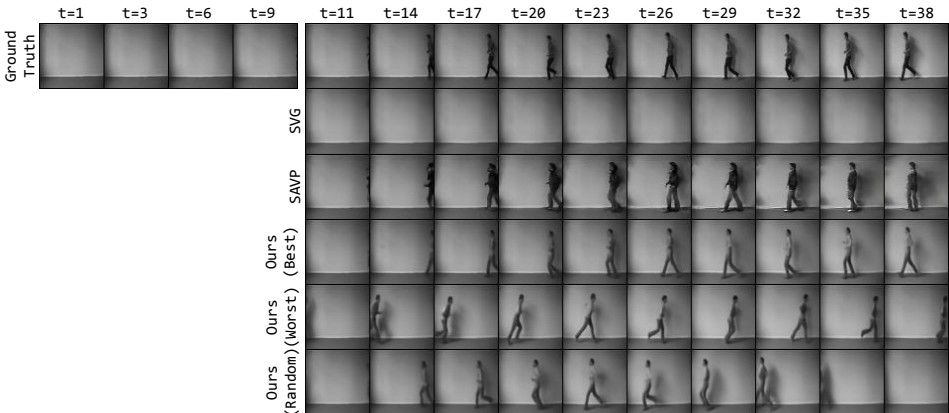

Figure 5: Conditioning frames and corresponding ground truth, best samples from SVG, SAVP and our method, and worst and random samples from our method, for an example of the KTH dataset. Samples are chosen according to their LPIPS with respect to the ground truth. SVG fails to make a person appear unlike SAVP and our model, and the latter better predicts the pose of the subject.

Figure 2b. We can see that our model better captures the dynamics of the problem compared to SVG as its performance decreases significantly less, even at a long-term horizon.

We also compare to two alternative versions of our model in Figure 2, where the residual dynamic function is replaced by an MLP or a GRU network (Cho et al., 2014). Our residual model outperforms both versions on the stochastic, and especially on the deterministic version of the dataset, showing its intrinsic advantage at modeling dynamics. Finally, on the deterministic version of Moving MNIST, we compare to an alternative where $z$ is entirely removed, resulting in a temporal model very close to the one presented in Chen et al. (2018). The loss of performance of this alternative model is significant, especially in SSIM, showing that our stochastic residual model offers a substantial advantage even when used in a deterministic environment.

**KTH Action dataset (KTH).** This dataset is composed of real-world videos of people performing a single action per video in front of different backgrounds (Schüldt et al., 2004). Uncertainty lies in the appearance of subjects, the actions they perform and how they are performed.

We outperform on this dataset every considered baseline for each metric, as depicted in Figure 4 and Table 2. In some videos, the subject only appears after the conditioning frames, requiring the model to sample the moment and location of the subject appearance, as well as its action. This critical case is illustrated in Figure 5. There, SVG fails to even generate a moving person; only SAVP and our model manage to do so, and our best sample is closer to the subject's poses compared to SAVP. Moreover, the worst and a random sample of our model demonstrate that it captures the diversity of the dataset by making a person appear at different time steps and with different speeds. An additional experiment on this dataset is included in Appendix G, studying the influence of the encoder and decoder architecture on SVG and our model.

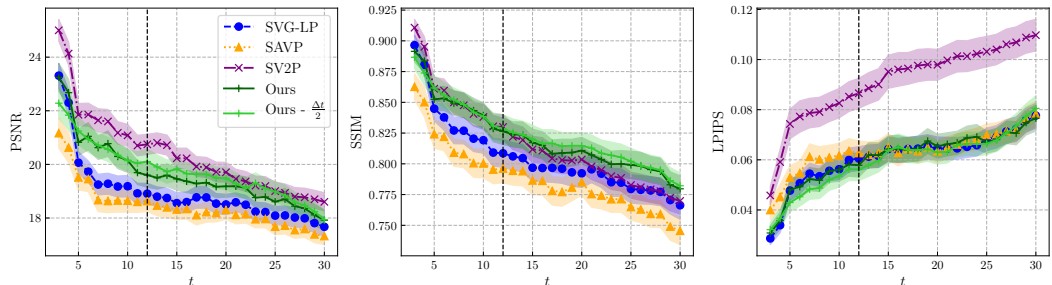

Figure 6: PSNR, SSIM and LPIPS scores with respect to $t$ for all tested models on the BAIR dataset.

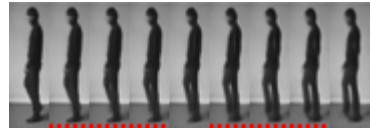

(a) Cropped KTH sample.

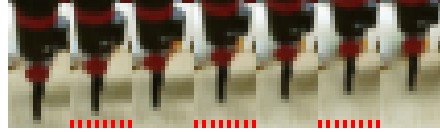

(b) Cropped BAIR sample.

Figure 7: Generation examples at doubled frame rate, using a halved $\Delta t$ compared to training. Frames including a bottom red dashed bar are intermediate frames.

Finally, Table 2 compares our method to its MLP and GRU alternative versions, leading to two conclusions. Firstly, it confirms the structural advantage of residual dynamics observed on Moving MNIST. On one hand, MLP better captures dynamics than GRU on KTH according to PSNR and SSIM, but loses in terms of realism according to LPIPS and FVD. On the other hand, the residual version shows a slight dynamics improvement with respect to both MLP and GRU, while substantially pushing further prediction realism. Secondly, all three versions of our model (residual, MLP, GRU) outperform prior methods. Therefore, this improvement is due to their common inference method, latent nature and content variable, strengthening our motivation to propose a non-autoregressive model.

**BAIR robot pushing dataset (BAIR).** This dataset contains videos of a Sawyer robotic arm pushing objects on a tabletop (Ebert et al., 2017). It is highly stochastic as the arm can change its direction at any moment. We achieve similar or better results compared to state-of-the-art models, as Figure 6 and Table 3 shows, and second-best PSNR behind SV2P, but the latter produces very blurry samples, which can be seen in Appendix H, yielding prohibitive LPIPS and FVD scores. In contrast, we achieve the highest SSIM overall, as well as state-of-the-art LPIPS and competitive FVD among these models. Note that we could not add VideoFlow to our experiments, due to the unavailability of pretrained models and numerical results. However, compared to PSNR, SSIM and LPIPS results reported by Kumar et al. (2019) for BAIR (the only tested dataset and metrics in their paper), our model appears to behave better than VideoFlow, which is on par with SAVP on these metrics.

**Varying frame rate in testing.** We challenge the ability of our model to use a different Euler step size than the one used in training (see Equation (2)). Figures 4 and 6 include corresponding results with a halved $\Delta t$. Prediction performances remain stable while generating twice as many frames (cf. Appendix F for further discussion). Our model is thus robust to the refinement of the Euler approximation, showing the quality of the learned dynamic which is close to continuous. In particular, this shows that our model learned a dynamic driven by a piecewise ODE, i.e., the learned dynamic of each interval between two consecutive frames is an ODE, as a constant $z$ is given on such interval. This can be used to generate frames at a higher frame rate than the training videos without supervision. We show in Figure 7 and Appendix F frames generated at a double and quadruple frame rate on BAIR and KTH. Both figures show smooth intermediate generated frames.

**Disentangling dynamics and content.** Let us show that the proposed model actually separates content from dynamics as discussed in Section 3.2. To this end, two sequences $\boldsymbol{x}^{\mathrm{s}}$ and $\boldsymbol{x}^{\mathrm{t}}$ are drawn

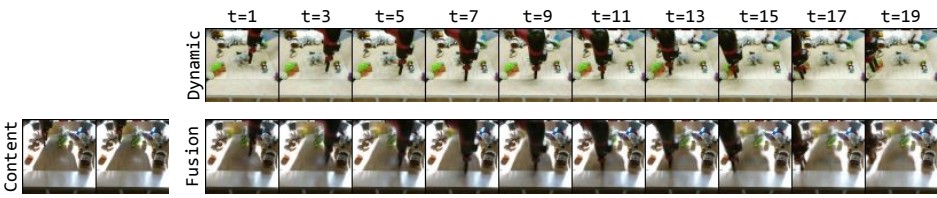

Figure 8: Video (bottom right) generated from the dynamic latent state $y$ inferred with a video (top) and the content variable $w$ computed with the conditioning frames of another video (bottom left). The generated video keeps the same background as the bottom left frames, while the robotic arm moves accordingly to the top frames.

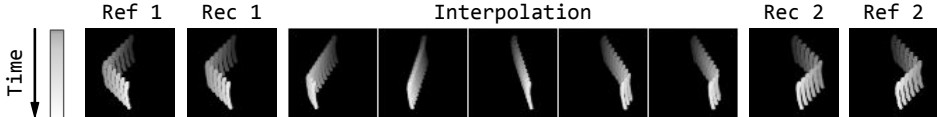

Figure 9: From left to right, $x^s$, $\widehat{x}^s$ (reconstruction of $x^s$ by the VAE of our model), results of the interpolation in the latent space between $x^s$ and $x^t$, $\widehat{x}^t$ and $x^t$. Each trajectory is materialized in shades of grey in the frames.

from the BAIR test set. While $x^s$ is used for extracting our content variable $w^s$, dynamic states $y^t$ are inferred with our model from $x^t$. New frame sequences $\widehat{x}$ are finally generated from the fusion of the content vector and the dynamics. This results in a content corresponding to the first sequence $x^s$ while moving according to the dynamics of the second sequence $x^t$, as observed in Figure 8. More samples for BAIR and KTH can be seen in Appendix H.

**Interpolation of dynamics.** Our state-space structure allows us to learn semantic representations in $y_t$. To highlight this feature, we test whether two Moving MNIST trajectories can be interpolated by linearly interpolating their inferred latent initial conditions. We begin by generating two trajectories $x^s$ and $x^t$ of a single moving digit. We infer their respective latent initial conditions $y_1^s$ and $y_1^t$. We then use our model to generate frame sequences from latent initial conditions linearly interpolated between $y_1^s$ and $y_1^t$. If it learned a meaningful latent space, the resulting trajectories should also be smooth interpolations between the directions of reference trajectories $x^s$ and $x^t$, and this is what we observe in Figure 9. Additional examples can be found in Appendix H.

## 5 CONCLUSION

We introduce a novel dynamic latent model for stochastic video prediction which, unlike prior image-autoregressive models, decouples frame synthesis and dynamics. This temporal model is based on residual updates of a small latent state that is showed to perform better than RNN-based models. This endows our method with several desirable properties, such as temporal efficiency and latent space interpretability. We experimentally demonstrate the performance and advantages of the proposed model, which outperforms prior state-of-the-art methods for stochastic video prediction. This work is, to the best of our knowledge, the first to propose a latent dynamic model scaling for video prediction. The proposed model is also novel with respect to the recent line of work dealing with neural networks and ODEs for temporal modeling; it is the first such residual model to scale to complex stochastic data such as videos.

We believe that the general principles of our model (state-space, residual dynamic, static content variable) can be generally applied to other models as well. Interesting future works include replacing the VRNN model of Minderer et al. (2019) in order to model the evolution of key-points, or leveraging the state-space nature of our model in model-based reinforcement learning.

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

## A  EVIDENCE LOWER BOUND

We develop in this section the computations of the variational lower bound for the proposed model.

Using the original variational lower bound of Kingma & Welling (2014) in Equation (8):

$$
\log p(\boldsymbol{x}_{1:T} \mid \boldsymbol{w})
$$
$$
\geq \mathbb{E}_{(\widetilde{\boldsymbol{z}}_{2:T}, \widetilde{\boldsymbol{y}}_{1:T}) \sim q_{Z,Y}} \log p(\boldsymbol{x}_{1:T} \mid \widetilde{\boldsymbol{z}}_{2:T}, \widetilde{\boldsymbol{y}}_{1:T}, \boldsymbol{w}) - D_{\mathrm{KL}}\big(q_{Z,Y} \,\big\|\, p(\boldsymbol{y}_{1:T}, \boldsymbol{z}_{2:T} \mid \boldsymbol{w})\big) \quad (8)
$$
$$
= \mathbb{E}_{(\widetilde{\boldsymbol{z}}_{2:T}, \widetilde{\boldsymbol{y}}_{1:T}) \sim q_{Z,Y}} \log p(\boldsymbol{x}_{1:T} \mid \widetilde{\boldsymbol{z}}_{2:T}, \widetilde{\boldsymbol{y}}_{1:T}, \boldsymbol{w}) - D_{\mathrm{KL}}\big(q(\boldsymbol{y}_1, \boldsymbol{z}_{2:T} \mid \boldsymbol{x}_{1:T}) \,\big\|\, p(\boldsymbol{y}_1, \boldsymbol{z}_{2:T})\big)
$$
$$
\tag{9}
$$
$$
= \mathbb{E}_{(\widetilde{\boldsymbol{z}}_{2:T}, \widetilde{\boldsymbol{y}}_{1:T}) \sim q_{Z,Y}} \sum_{t=1}^{T} \log p(\boldsymbol{x}_t \mid \widetilde{\boldsymbol{y}}_t, \boldsymbol{w}) - D_{\mathrm{KL}}\big(q(\boldsymbol{y}_1, \boldsymbol{z}_{2:T} \mid \boldsymbol{x}_{1:T}) \,\big\|\, p(\boldsymbol{y}_1, \boldsymbol{z}_{2:T})\big), \quad (10)
$$

where:

- Equation (9) is given by the forward and inference models factorizing $p$ and $q$ in Equations (4) and (5) and illustrated by, respectively, Figures 1a and 1b:
  - the $\boldsymbol{z}$ variables and $\boldsymbol{y}_1$ are independent from $\boldsymbol{w}$ to $p$ and $q$;
  - the $\boldsymbol{y}_{2:T}$ variables are deterministic functions of $\boldsymbol{y}_1$ and $\boldsymbol{z}_{2:T}$ with respect to $p$ and $q$;
- Equation (10) results from the factorization of $p(\boldsymbol{x}_{1:T} \mid \boldsymbol{y}_{1:T}, \boldsymbol{z}_{1:T}, \boldsymbol{w})$ in Equation (4).

From there, by using the integral formulation of $D_{\mathrm{KL}}$:

$$\log p(\boldsymbol{x}_{1:T} \mid \boldsymbol{w})$$

$$\geq \mathbb{E}_{(\widetilde{\boldsymbol{z}}_{2:T}, \widetilde{\boldsymbol{y}}_{1:T}) \sim q_{Z,Y}} \sum_{t=1}^{T} \log p(\boldsymbol{x}_t \mid \widetilde{\boldsymbol{y}}_t, \boldsymbol{w})$$

$$+ \int \cdots \int_{\boldsymbol{y}_1, \boldsymbol{z}_{2:T}} q(\boldsymbol{y}_1, \boldsymbol{z}_{2:T} \mid \boldsymbol{x}_{1:T}) \log \frac{p(\boldsymbol{y}_1, \boldsymbol{z}_{2:T})}{q(\boldsymbol{y}_1, \boldsymbol{z}_{2:T} \mid \boldsymbol{x}_{1:T})} \, \mathrm{d}\boldsymbol{z}_{2:T} \, \mathrm{d}\boldsymbol{y}_1 \tag{11}$$

$$= \mathbb{E}_{(\widetilde{\boldsymbol{z}}_{2:T}, \widetilde{\boldsymbol{y}}_{1:T}) \sim q_{Z,Y}} \sum_{t=1}^{T} \log p(\boldsymbol{x}_t \mid \widetilde{\boldsymbol{y}}_t, \boldsymbol{w}) - D_{\mathrm{KL}}\big( q(\boldsymbol{y}_1 \mid \boldsymbol{x}_{1:T}) \,\big\|\, p(\boldsymbol{y}_1) \big)$$

$$+ \mathbb{E}_{\widetilde{\boldsymbol{y}}_1 \sim q(\boldsymbol{y}_1 \mid \boldsymbol{x}_{1:T})} \left[ \int \cdots \int_{\boldsymbol{z}_{2:T}} q(\boldsymbol{z}_{2:T} \mid \boldsymbol{x}_{1:T}, \widetilde{\boldsymbol{y}}_1) \log \frac{p(\boldsymbol{z}_{2:T} \mid \widetilde{\boldsymbol{y}}_1)}{q(\boldsymbol{z}_{2:T} \mid \boldsymbol{x}_{1:T}, \widetilde{\boldsymbol{y}}_1)} \, \mathrm{d}\boldsymbol{z}_{2:T} \right] \tag{12}$$

$$= \mathbb{E}_{(\widetilde{\boldsymbol{z}}_{2:T}, \widetilde{\boldsymbol{y}}_{1:T}) \sim q_{Z,Y}} \sum_{t=1}^{T} \log p(\boldsymbol{x}_t \mid \widetilde{\boldsymbol{y}}_t, \boldsymbol{w}) - D_{\mathrm{KL}}\big( q(\boldsymbol{y}_1 \mid \boldsymbol{x}_{1:k}) \,\big\|\, p(\boldsymbol{y}_1) \big)$$

$$+ \mathbb{E}_{\widetilde{\boldsymbol{y}}_1 \sim q(\boldsymbol{y}_1 \mid \boldsymbol{x}_{1:k})} \left[ \int \cdots \int_{\boldsymbol{z}_{2:T}} q(\boldsymbol{z}_{2:T} \mid \boldsymbol{x}_{1:T}, \widetilde{\boldsymbol{y}}_1) \log \frac{p(\boldsymbol{z}_{2:T} \mid \widetilde{\boldsymbol{y}}_1)}{q(\boldsymbol{z}_{2:T} \mid \boldsymbol{x}_{1:T}, \widetilde{\boldsymbol{y}}_1)} \, \mathrm{d}\boldsymbol{z}_{2:T} \right] \tag{13}$$

$$= \mathbb{E}_{(\widetilde{\boldsymbol{z}}_{2:T}, \widetilde{\boldsymbol{y}}_{1:T}) \sim q_{Z,Y}} \sum_{t=1}^{T} \log p(\boldsymbol{x}_t \mid \widetilde{\boldsymbol{y}}_t, \boldsymbol{w}) - D_{\mathrm{KL}}\big( q(\boldsymbol{y}_1 \mid \boldsymbol{x}_{1:k}) \,\big\|\, p(\boldsymbol{y}_1) \big)$$

$$+ \mathbb{E}_{\widetilde{\boldsymbol{y}}_1 \sim q(\boldsymbol{y}_1 \mid \boldsymbol{x}_{1:k})} \left[ \int \cdots \int_{\boldsymbol{z}_{2:T}} \prod_{t=2}^{T} q(\boldsymbol{z}_t \mid \boldsymbol{x}_{1:t}) \sum_{t=2}^{T} \log \frac{p(\boldsymbol{z}_t \mid \widetilde{\boldsymbol{y}}_1, \boldsymbol{z}_{2:t-1})}{q(\boldsymbol{z}_t \mid \boldsymbol{x}_{1:t})} \, \mathrm{d}\boldsymbol{z}_{2:T} \right] \tag{14}$$

$$= \mathbb{E}_{(\widetilde{\boldsymbol{z}}_{2:T}, \widetilde{\boldsymbol{y}}_{1:T}) \sim q_{Z,Y}} \sum_{t=1}^{T} \log p(\boldsymbol{x}_t \mid \widetilde{\boldsymbol{y}}_t, \boldsymbol{w}) - D_{\mathrm{KL}}\big( q(\boldsymbol{y}_1 \mid \boldsymbol{x}_{1:k}) \,\big\|\, p(\boldsymbol{y}_1) \big)$$

$$- \mathbb{E}_{\widetilde{\boldsymbol{y}}_1 \sim q(\boldsymbol{y}_1 \mid \boldsymbol{x}_{1:k})} D_{\mathrm{KL}}\big( q(\boldsymbol{z}_2 \mid \boldsymbol{x}_{1:t}) \,\big\|\, p(\boldsymbol{z}_2 \mid \widetilde{\boldsymbol{y}}_1) \big)$$

$$+ \mathbb{E}_{\widetilde{\boldsymbol{y}}_1 \sim q(\boldsymbol{y}_1 \mid \boldsymbol{x}_{1:k})} \mathbb{E}_{\widetilde{\boldsymbol{z}}_2 \sim q(\boldsymbol{z}_2 \mid \boldsymbol{x}_{1:2})}$$

$$\left[ \int \cdots \int_{\boldsymbol{z}_{3:T}} \prod_{t=3}^{T} q(\boldsymbol{z}_t \mid \boldsymbol{x}_{1:t}) \sum_{t=3}^{T} \log \frac{p(\boldsymbol{z}_t \mid \boldsymbol{y}_1, \widetilde{\boldsymbol{z}}_{2:t-1})}{q(\boldsymbol{z}_t \mid \boldsymbol{x}_{1:t})} \, \mathrm{d}\boldsymbol{z}_{3:T} \right], \tag{15}$$

where:

- Equation (13) follows from the inference model of Equation (5), where $\boldsymbol{y}_1$ only depends on $\boldsymbol{x}_{1:k}$;
- Equation (14) is obtained from the factorizations of Equations (4) and (5).

By iterating Equation (15)'s step on $\boldsymbol{z}_3, \ldots, \boldsymbol{z}_T$ and factorizing all expectations, we obtain:

$$\tag{16}$$

$$\log p(\boldsymbol{x}_{1:T} \mid \boldsymbol{w})$$

$$\geq \mathbb{E}_{(\widetilde{\boldsymbol{z}}_{2:T}, \widetilde{\boldsymbol{y}}_{1:T}) \sim q_{Z,Y}} \sum_{t=1}^{T} \log p(\boldsymbol{x}_t \mid \widetilde{\boldsymbol{y}}_t, \boldsymbol{w}) - D_{\mathrm{KL}}\big(q(\boldsymbol{y}_1 \mid \boldsymbol{x}_{1:k}) \, \big\| \, p(\boldsymbol{y}_1)\big)$$

$$- \mathbb{E}_{\widetilde{\boldsymbol{y}}_1 \sim q(\boldsymbol{y}_1 \mid \boldsymbol{x}_c)}\Big(\mathbb{E}_{\widetilde{\boldsymbol{z}}_t \sim q(\boldsymbol{z}_t \mid \boldsymbol{x}_{1:t})}\Big)_{t=2}^{T} \sum_{t=2}^{T} D_{\mathrm{KL}}\big(q(\boldsymbol{z}_t \mid \boldsymbol{x}_{1:t}) \, \big\| \, p(\boldsymbol{z}_t \mid \widetilde{\boldsymbol{y}}_1, \widetilde{\boldsymbol{z}}_{1:t-1})\big),$$

$$(17)$$

and we finally retrieve Equation (6) by using the factorization of Equation (5):

$$\log p(\boldsymbol{x}_{1:T} \mid \boldsymbol{w})$$

$$\geq \mathbb{E}_{(\widetilde{\boldsymbol{z}}_{2:T}, \widetilde{\boldsymbol{y}}_{1:T}) \sim q_{Z,Y}} \sum_{t=1}^{T} \log p(\boldsymbol{x}_t \mid \widetilde{\boldsymbol{y}}_t, \boldsymbol{w}) - D_{\mathrm{KL}}\big(q(\boldsymbol{y}_1 \mid \boldsymbol{x}_{1:k}) \, \big\| \, p(\boldsymbol{y}_1)\big)$$

$$- \mathbb{E}_{(\widetilde{\boldsymbol{z}}_{2:T}, \widetilde{\boldsymbol{y}}_{1:T}) \sim q_{Z,Y}} \sum_{t=2}^{T} D_{\mathrm{KL}}\big(q(\boldsymbol{z}_t \mid \boldsymbol{x}_{1:t}) \, \big\| \, p(\boldsymbol{z}_t \mid \widetilde{\boldsymbol{y}}_{t-1})\big).$$

$$(18)$$

# B  DATASETS DETAILS

## B.1  STOCHASTIC MOVING MNIST (SM-MNIST)

This dataset consists in one or two train MNIST digits (LeCun et al., 1998) of size $27 \times 27$ moving linearly within a $64 \times 64$ frame and randomly bounce against its border, sampling a new direction and velocity at each bounce (Denton & Fergus, 2018). We use the same settings as Denton & Fergus (2018), train all models on 15 timesteps and condition them at test time on 5 frames. Note that we adapted the dataset to sample more coherent bounces: the original dataset computes digit trajectories that are dependent on the chosen framerate, unlike our corrected version of the dataset. We consequently retrained SVG on this dataset, obtaining comparable results as those originally presented by Denton & Fergus (2018). Test data were produced by generating 5000 samples with a different digit for each sequence coming from the MNIST test set.

## B.2  KTH ACTION DATASET (KTH)

This dataset is composed of real-world $64 \times 64$ videos of 25 people performing one of six actions (walking, jogging, running, boxing, handwaving and handclapping) in front of different backgrounds (Schüldt et al., 2004). Uncertainty lies in the appearance of subjects, the action they perform and how it is performed. The training set is formed with actions from 20 people, the remaining five being used for testing. Training is performed by sampling sub-sequences of size 20 in the train set. The test set is composed of 1000 randomly sampled sub-sequences of size 40.

## B.3  BAIR ROBOT PUSHING DATASET (BAIR)

This dataset contains $64 \times 64$ videos of a Sawyer robotic arm pushing objects on a tabletop (Ebert et al., 2017). It is highly stochastic as the arm can change its direction at any moment. Training is performed on 12 frames and testing is done with two conditioning frames on the provided test set, consisting of 256 sequences of 30 frames.

# C  TRAINING DETAILS

## C.1  SPECIFICATIONS

We used Python 3.7.4 and PyTorch 1.2.0 (Paszke et al., 2017) to implement our model. Each model was trained on a Nvidia GPUs with CUDA 10 in mixed-precision training with the help of Apex.[2]

## C.2  ARCHITECTURE

**Encoder and decoder architecture.**  Both $g_\theta$ and $h_\phi$ are chosen to have different architectures depending on the dataset. We used the same architectures as in Denton & Fergus (2018): a DCGAN

---

[2]https://github.com/nvidia/apex.

discriminator and generator architecture (Radford et al., 2016) for Moving MNIST, and a VGG16 (Simonyan & Zisserman, 2015) architecture (mirrored for $h_\phi$) for BAIR and KTH. In both cases, the output of $h_\phi$ (i.e., $\widetilde{x}$) is a vector of size 128, and $g_\theta$ and $h_\phi$ weights are initialized using a centered normal distribution with a standard deviation of 0.02.

For the Moving MNIST dataset, the content variable $w$ is obtained directly from $\widetilde{x}$ and is thus a vector of size 128. For KTH and BAIR, we supplement this vectorial variable with skip connections from all layers of the encoder $g_\theta$ that are then fed to the decoder $h_\phi$ to handle complex backgrounds. For Moving MNIST, the number of frames $k$ used to compute the content variable is 5; for KTH, it is 3; for BAIR, it is 2.

**LSTM architecture.** The LSTM used for all datasets has a single layer of LSTM cells with a hidden state size of 256.

**MLP architecture.** All MLPs used in inference (with parameters $\phi$) have three linear layers with hidden size 256 and leaky ReLU activations. All MLPs used in the forward model (with parameters $\theta$) have four linear layers with hidden size 512 and leaky ReLU activations. Weights of $f_\theta$, in particular, are orthogonally initialized with a gain of 1.41, while the other MLPs are initialized with default weight initialization of PyTorch.

**Sizes of latent variables.** The sizes of the latent variables in our model are the following: for Moving MNIST, $y$ and $z$ have size 20; for KTH and BAIR, $y$ and $z$ have size 50.

**Euler step size** All models but those trained on KTH are trained with $\Delta t = 1$. Models on KTH are trained with $\Delta t = \frac{1}{2}$.

### C.3 OPTIMIZATION

**Loss function.** All models are trained using the Adam optimizer (Kingma & Ba, 2015) with learning rate $3 \times 10^{-4}$ and $\lambda = 1$. The batch size for Moving MNIST and BAIR is chosen to be 128, and the batch size for KTH is chosen to be 100.

Following (Higgins et al., 2017), we use $\beta = 1$ (cf. Equation (7)), except for the Moving MNIST dataset where the $\beta$ factor in front of the KL on $z$ (last term of Equation (6)) is equal to 2.

**Variance of the observation.** The variance $\nu$ used in the observation probability distribution $\mathcal{G}(g_\theta(y)) = \mathcal{N}(g_\theta(y), \nu I)$ is chosen as follows:

- for Moving MNIST, $\nu = 1$;
- for KTH, $\nu = 4 \times 10^{-2}$;
- for BAIR, $\nu = \frac{1}{2}$.

**Number of optimization steps.** The number of optimization steps is the following for the different datasets:

- Moving MNIST (stochastic): $1\,000\,000$ steps with additional $100\,000$ steps where the learning rated is linearly decreased to 0;
- Moving MNIST (deterministic): $800\,000$ steps with additional $100\,000$ steps where the learning rated is linearly decreased to 0;
- KTH: $200\,000$ steps, the final model being chosen among several checkpoints as the one having the best evaluation score (which differs from the test score as we extract from the train set an evaluation set);
- BAIR: $250\,000$ steps, the final model is chosen as for KTH.

Table 1: Numerical results (mean and 95%-confidence interval) for PSNR and SSIM for tested methods on the two-digits Moving MMNIST dataset. Bold scores indicate the best performing method and, where appropriate, scores whose means lie in the confidence interval of the best performing method.

| Models | Stochastic | | Deterministic | |
|--------|------------|------|---------------|------|
| | PSNR | SSIM | PSNR | SSIM |
| SVG | $14.45 \pm 0.06$ | $0.7070 \pm 0.0021$ | $12.93 \pm 0.05$ | $0.6245 \pm 0.0022$ |
| Ours | $\mathbf{16.90 \pm 0.09}$ | $\mathbf{0.7789 \pm 0.0025}$ | $\mathbf{16.49 \pm 0.06}$ | $\mathbf{0.7808 \pm 0.0020}$ |
| Ours - MLP | $16.55 \pm 0.09$ | $0.7693 \pm 0.0024$ | $14.32 \pm 0.06$ | $0.6895 \pm 0.0023$ |
| Ours - GRU | $15.81 \pm 0.08$ | $0.7463 \pm 0.0023$ | $13.16 \pm 0.05$ | $0.6318 \pm 0.0022$ |

Table 2: Numerical results (mean and 95%-confidence interval, when relevant) for PSNR, SSIM, LPIPS, and FVD for tested methods on the KTH dataset. Bold scores indicate the best performing method for each metric and, where appropriate, scores whose means lie in the confidence interval of the best performing method.

| Models | PSNR | SSIM | LPIPS | FVD |
|--------|------|------|-------|-----|
| SV2P | $28.18 \pm 0.39$ | $0.8141 \pm 0.0068$ | $0.2049 \pm 0.0080$ | 636 |
| SAVP | $26.51 \pm 0.36$ | $0.7560 \pm 0.0083$ | $0.1120 \pm 0.0058$ | 376 |
| SVG | $26.99 \pm 0.33$ | $0.8291 \pm 0.0074$ | $0.1083 \pm 0.0058$ | 265 |
| Ours | $\mathbf{29.69 \pm 0.37}$ | $\mathbf{0.8697 \pm 0.0057}$ | $\mathbf{0.0736 \pm 0.0036}$ | $\mathbf{224}$ |
| Ours - GRU | $29.13 \pm 0.38$ | $0.8590 \pm 0.0060$ | $0.0790 \pm 0.0039$ | 237 |
| Ours - MLP | $\mathbf{29.49 \pm 0.38}$ | $0.8626 \pm 0.0061$ | $0.0825 \pm 0.0042$ | 290 |

## D  ADDITIONAL NUMERICAL RESULTS

Tables 1 to 3 present, respectively, numerical results for PSNR, SSIM and LPIPS averaged over all time steps for our methods and considered baselines on the SM-MNIST, KTH and BAIR datasets, corresponding to Figures 2, 4 and 6.

## E  PENDULUM EXPERIMENTS

We test the ability of our model to model the dynamics of a common dataset used in the literature of state-space models (Karl et al., 2017; Fraccaro et al., 2017), Pendulum (Karl et al., 2017). It consists of noisy observations of a dynamic torque-controlled pendulum; it is stochastic as the information of this control is not available. We test our model, without the content variable $w$, in the same setting as DVBF (Karl et al., 2017) and KVAE (Fraccaro et al., 2017) and report the corresponding ELBO scores in Table 4. The encoders and decoders for all methods are MLPs.

Our model outperforms DVBF and is merely beaten by KVAE. This can be explained by the nature of the KVAE model, whose sequential model is not learned using a VAE but a Kalman filter allowing

Table 3: Numerical results (mean and 95%-confidence interval, when relevant) with respect to PSNR, SSIM, LPIPS, and FVD for tested methods on the BAIR dataset. Bold scores indicate the best performing method for each metric and, where appropriate, scores whose means lie in the confidence interval of the best performing method.

| Models | PSNR | SSIM | LPIPS | FVD |
|--------|------|------|-------|-----|
| SV2P | $\mathbf{20.39 \pm 0.42}$ | $\mathbf{0.8169 \pm 0.0110}$ | $0.0912 \pm 0.0063$ | 955 |
| SAVP | $18.44 \pm 0.40$ | $0.7886 \pm 0.0117$ | $0.0634 \pm 0.0048$ | $\mathbf{152}$ |
| SVG | $18.95 \pm 0.41$ | $0.8057 \pm 0.0116$ | $\mathbf{0.0609 \pm 0.0046}$ | 253 |
| Ours | $19.64 \pm 0.45$ | $\mathbf{0.8211 \pm 0.0110}$ | $0.0610 \pm 0.0048$ | 181 |

Table 4: ELBO score for DVBF, KVAE and our model on the Pendulum dataset. The bold score indicates the best performing method.

| Score | DVBF | KVAE | Ours |
|-------|------|------|------|
| ELBO | 798.56 | **807.02** | 806.12 |

Table 5: Numerical results for PSNR, SSIM, and LPIPS on BAIR of our model trained with $\Delta t = 1$ and tested with different values of $\Delta t$.

| Step size $\Delta t$ | PSNR | SSIM | LPIPS |
|----------------------|------|------|-------|
| $\boldsymbol{\Delta t = 1}$ | $19.64 \pm 0.45$ | $0.8210 \pm 0.0110$ | $0.0612 \pm 0.0048$ |
| $\Delta t = \frac{1}{2}$ | $19.76 \pm 0.44$ | $0.8235 \pm 0.0110$ | $0.0597 \pm 0.0047$ |
| $\Delta t = \frac{1}{3}$ | $19.82 \pm 0.45$ | $0.8245 \pm 0.0111$ | $0.0593 \pm 0.0048$ |
| $\Delta t = \frac{1}{4}$ | $19.83 \pm 0.46$ | $0.8242 \pm 0.0111$ | $0.0593 \pm 0.0049$ |
| $\Delta t = \frac{1}{5}$ | $19.85 \pm 0.46$ | $0.8243 \pm 0.0111$ | $0.0591 \pm 0.0048$ |

exact inference in the latent space. On the contrary, DVBF is learned, like our model, by a sequential VAE, and is thus much closer to our model than KVAE. This result then shows that the dynamic model that we chose in the context of sequential VAEs is more adapted on this dataset than the one of DVBF, and achieve results close to a method taking advantage of exact inference using adapted tools such as Kalman filters.

## F   INFLUENCE OF THE EULER STEP SIZE

Table 5 details the numerical results of our model trained on BAIR with $\Delta t = 1$ and tested with different values of $\Delta t$. It shows that, when refining the Euler approximation, our model can improve its performance in a setting that is unseen during training. Results stabilize when $\Delta t$ is small enough, showing that the model is close to the continuous limit.

Tables 6 and 7 details the numerical results of our model trained on KTH with, respectively, $\Delta t = 1$ and $\Delta t = \frac{1}{2}$, and tested with different values of $\Delta t$. They show that if $\Delta t$ is chosen too high when training (here, $\Delta t = 1$), the model drops in performance when refining the Euler approximation. We assume that this phenomenon arises because the Euler approximation used in training is too rough, making the model adapt to a very discretized dynamic that cannot be transferred to smaller Euler step sizes. Indeed, when training with smaller step size, (here, $\Delta t = \frac{1}{2}$), results in the training settings are equivalent while results obtained with a lower $\Delta t$ are now much closer, if not equivalent, to the nominal ones. This shows that the model learned a continuous dynamic if learned with a small enough step size.

Note that the loss of performance using a higher $\Delta t$ in testing than in training, like in Table 7, is expected as it corresponds to loosening the Euler approximation compared to training. However, even in this adversarial setting, our model maintains state-of-the-art results, demonstrating the quality of the learned dynamic as it can be further discretized if needed at the cost of a reasonable drop in performance.

Table 6: Numerical resuls for PSNR, SSIM, and LPIPS on KTH of our model trained with $\Delta t = 1$ and tested with different values of $\Delta t$.

| Step size $\Delta t$ | PSNR | SSIM | LPIPS |
|----------------------|------|------|-------|
| $\boldsymbol{\Delta t = 1}$ | $29.76 \pm 0.38$ | $0.8681 \pm 0.0057$ | $0.0737 \pm 0.0057$ |
| $\Delta t = \frac{1}{2}$ | $29.05 \pm 0.42$ | $0.8539 \pm 0.0066$ | $0.0882 \pm 0.0050$ |
| $\Delta t = \frac{1}{3}$ | $29.05 \pm 0.42$ | $0.8509 \pm 0.0069$ | $0.0924 \pm 0.0055$ |
| $\Delta t = \frac{1}{4}$ | $28.98 \pm 0.42$ | $0.8496 \pm 0.0069$ | $0.0939 \pm 0.0056$ |
| $\Delta t = \frac{1}{5}$ | $28.95 \pm 0.42$ | $0.8490 \pm 0.0070$ | $0.0948 \pm 0.0057$ |

Table 7: Numerical results for PSNR, SSIM, and LPIPS on KTH of our model trained with $\Delta t = \frac{1}{2}$ and tested with different values of $\Delta t$.

| Step size $\Delta t$ | PSNR | SSIM | LPIPS |
|---|---|---|---|
| $\Delta t = 1$ | $28.80 \pm 0.41$ | $0.8495 \pm 0.0068$ | $0.0994 \pm 0.0057$ |
| $\boldsymbol{\Delta t = \frac{1}{2}}$ | $29.69 \pm 0.37$ | $0.8697 \pm 0.0057$ | $0.0736 \pm 0.0036$ |
| $\Delta t = \frac{1}{3}$ | $29.52 \pm 0.38$ | $0.8656 \pm 0.0059$ | $0.0777 \pm 0.0041$ |
| $\Delta t = \frac{1}{4}$ | $29.43 \pm 0.39$ | $0.8633 \pm 0.0061$ | $0.0790 \pm 0.0042$ |
| $\Delta t = \frac{1}{5}$ | $29.35 \pm 0.39$ | $0.8615 \pm 0.0062$ | $0.0810 \pm 0.0045$ |

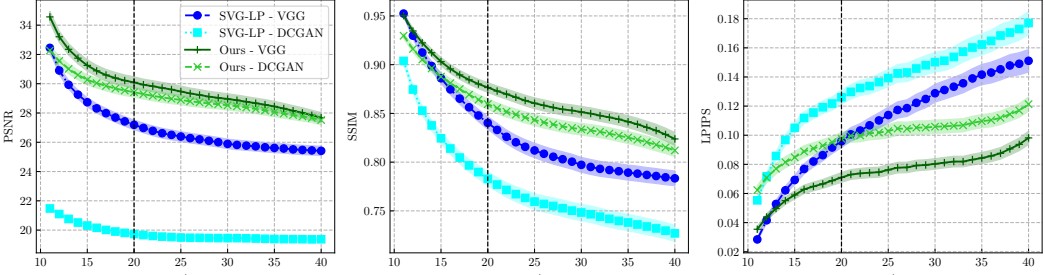

Figure 10: PNSR, SSIM and LPIPS scores with respect to $t$ on the KTH dataset for SVG and our model with two choices of encoder and decoder architecture for each: DCGAN and VGG.

## G  AUTOREGRESSIVITY AND IMPACT OF ENCODER AND DECODER ARCHITECTURE

Figure 10 exposes the numerical results on KTH of our model and SVG for different choices of architectures: DCGAN and VGG.

Since DCGAN is a less powerful architecture than VGG, results of each method with VGG are expectedly better than those of the same method with DCGAN. Moreover, our model outperforms SVG for any fixed choice of encoder and decoder architecture, which is coherent with Figure 4.

We observe, however, that the difference between a method using VGG and its DCGAN counterpart differs depending on the model. Ours shows more robustness to changing of encoder and decoder architecture, as it loses much less performance than SVG when switching to a less powerful architecture. Indeed, while the difference in LPIPS is similar for both models (as expected from a score evaluating the realism of produced frames), the loss of SVG is significantly larger than our loss in terms of SSIM, and in particular PNSR. This shows that reducing the capacity of the encoders and decoders of SVG not only hurts its ability to produce realistic frames as expected but also substantially lowers its ability to learn a good dynamic. We assume that this phenomenon is caused by the autoregressive nature of SVG, which makes it reliant of the performance of its encoders and decoders. This supports our motivation to propose a non-autoregressive model for stochastic video prediction.

## H  ADDITIONAL SAMPLES

This section includes some additional samples corresponding to experiments described in Section 4.

### H.1  STOCHASTIC MOVING MNIST

We present in Figures 11 to 14 additional samples from SVG and our model on SM-MNIST.

In particular, Figure 13 shows SVG changing a digit shape in the course of a prediction even though it does not cross another digit, whereas ours maintain the digit shape. We assume that this advantage of ours comes from the latent nature of the dynamic of our model and the use in our of a static content variable that is prevented from containing temporal information. Indeed, even when the best sample

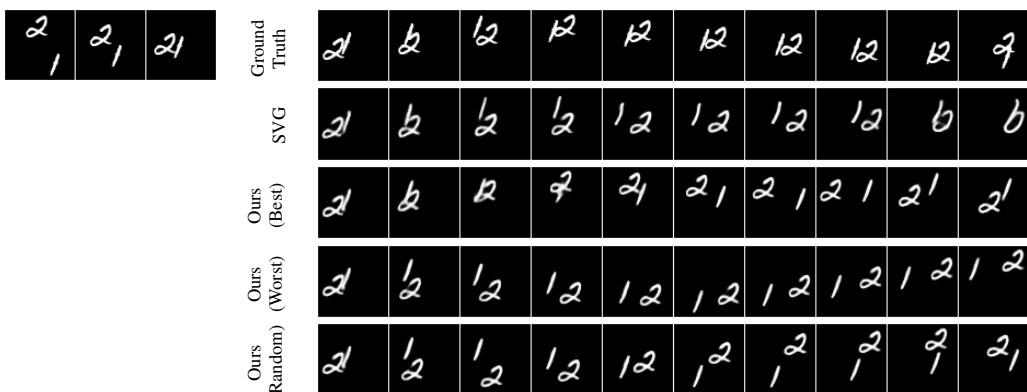

Figure 11: Conditioning frames and corresponding ground truth and best samples with respect to PSNR from SVG and our method, and worst and random samples from our method, for an example of the SM-MNIST dataset.

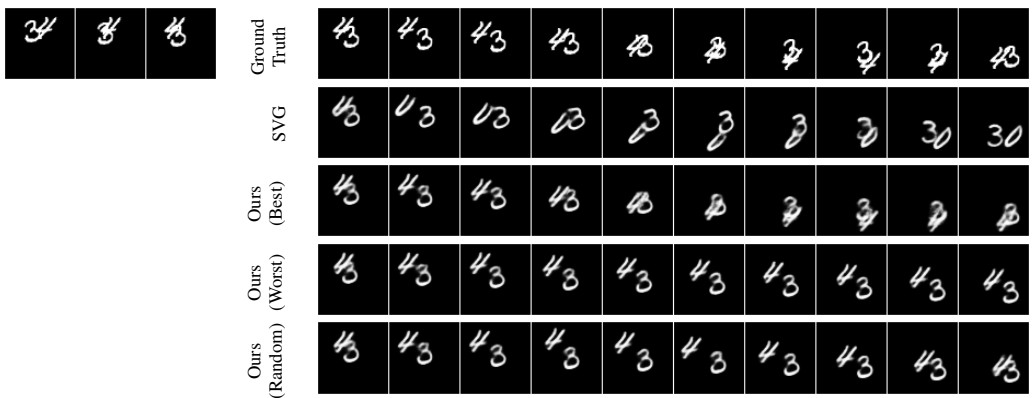

Figure 12: Additional samples for the SM-MNIST dataset (cf. Figure 11).

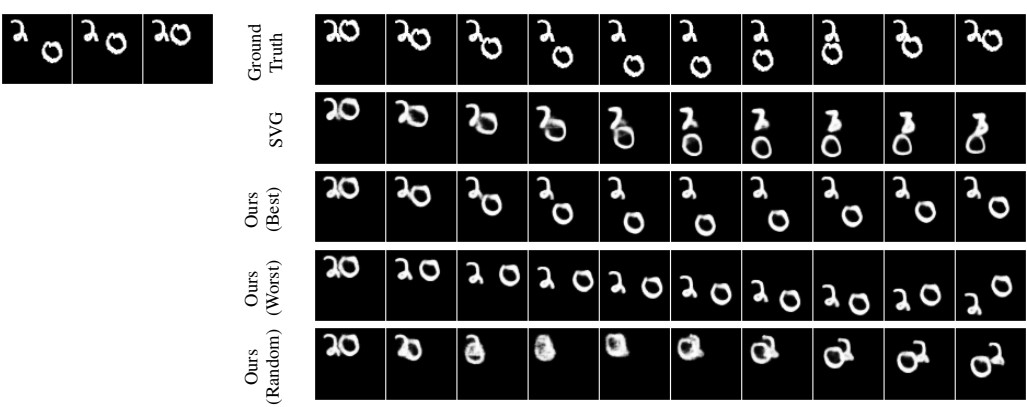

Figure 13: Additional samples for the SM-MNIST dataset (cf. Figure 11). SVG fails to maintain the shape of a digit, while ours is temporally coherent.

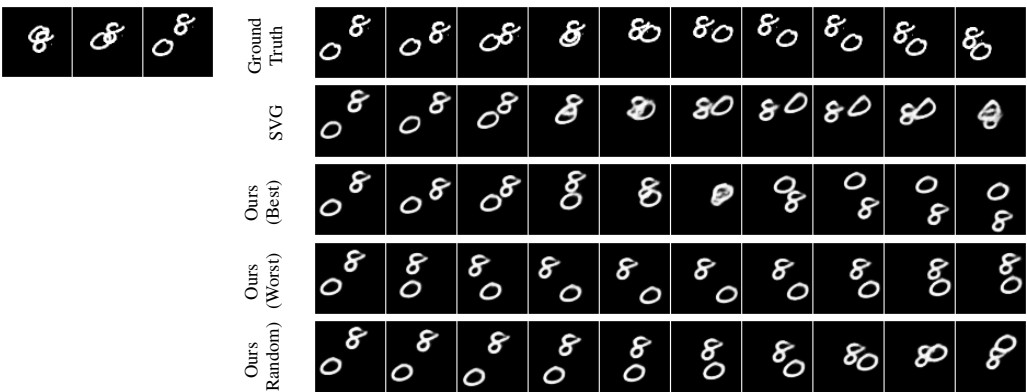

Figure 14: Additional samples for the SM-MNIST dataset (cf. Figure 11). This example was chosen in the worst $1\%$ test examples of our model with respect to PSNR. Despite this adversarial criterion, our model maintains temporal consistency as digits are not deformed in the course of the video.

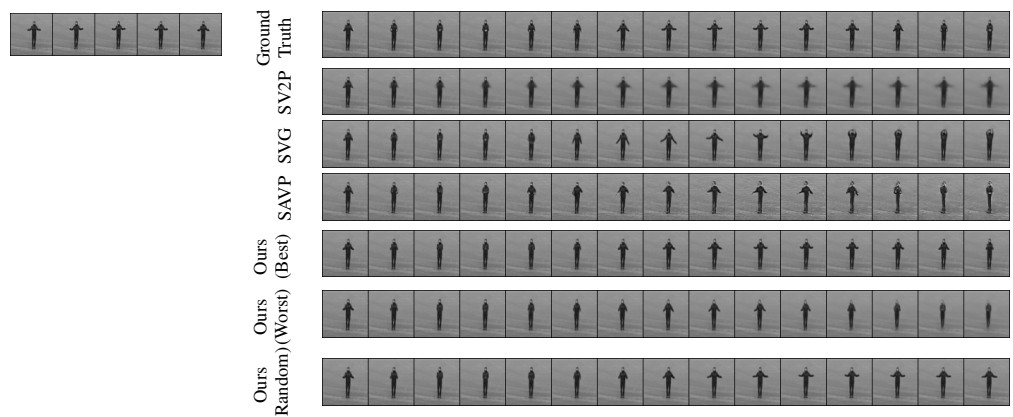

Figure 15: Conditioning frames and corresponding ground truth, best samples from SVG, SAVP and our method, and worst and random samples from our method, for an example of the KTH dataset. Samples are chosen according to their LPIPS with respect to the ground truth. On this specific task (clapping), all methods but SV2P (which produce blurry predictions) perform well, even though ours stays closer to the ground truth.

from our model is not close from the ground truth of the dataset, like in Figure 14, the shapes of the digits are still maintained by our model.

## H.2 KTH

We present in Figures 15 to 19 additional samples from SV2P, SVG, SAVP and our model on KTH, with additional insights.

## H.3 BAIR

We present in Figures 20 to 22 additional samples from SV2P, SVG, SAVP and our model on BAIR, with additional insights.

## H.4 OVERSAMPLING

We present in Figure 23 additional examples of video generation at a doubled frame rate by our model.

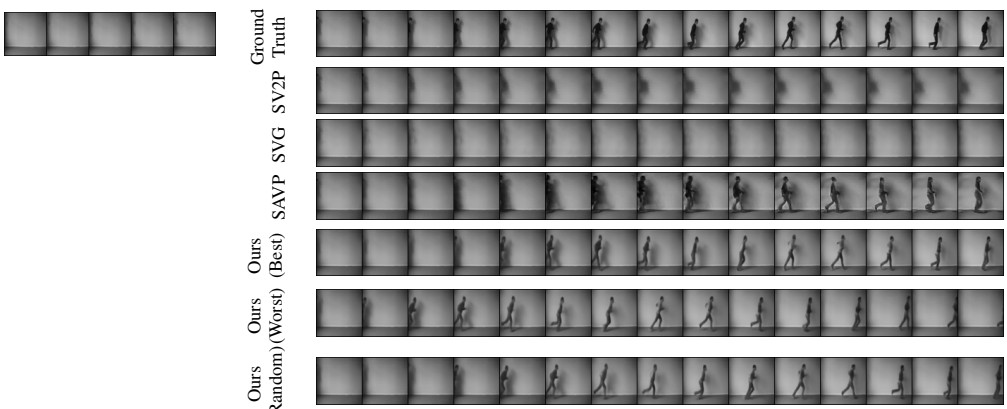

Figure 16: Additonal samples for the KTH dataset (cf. Figure 15). In this example, the shadow of the subject is visible in the last conditioning frames, foreshadowing its appearance. This is a failure case for SVG and SAVP which only produce an indistinct shadow, whereas SAVP and our model make the subject appear. Yet, SAVP produces the wrong action and an inconsistent subject in its best sample, while ours is correct.

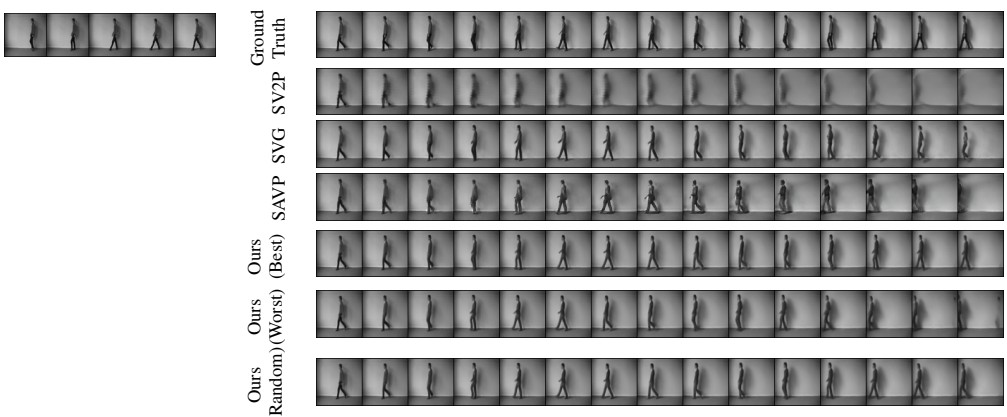

Figure 17: Additonal samples for the KTH dataset (cf. Figure 15). This example is a failure case for each method: SV2P produce blurry frames, SVG and SAVP are not consistent (change of action or subject appearance in the video), and our model produces a ghost image at the end of the prediction on the worst sample only.

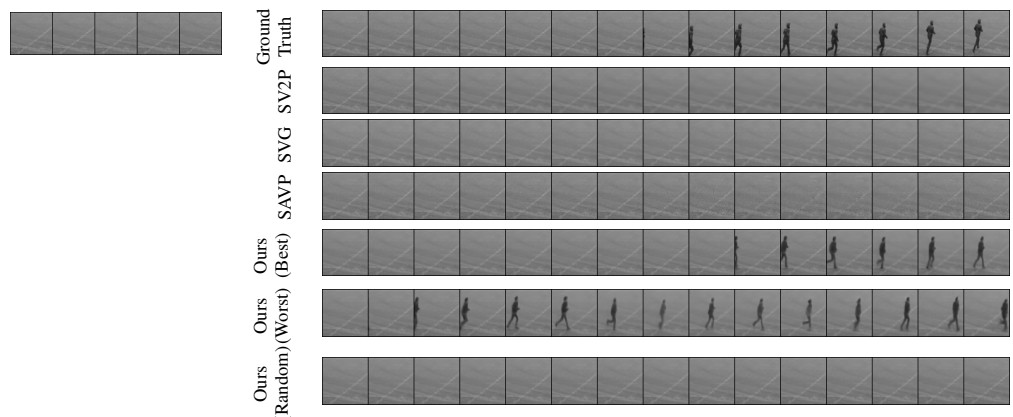

Figure 18: Additonal samples for the KTH dataset (cf. Figure 15). Our model is the only one to make a subject appear in the ground truth.

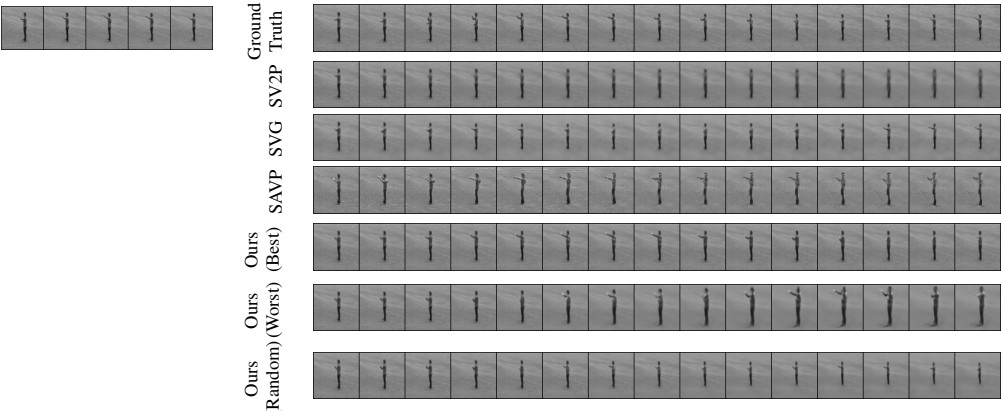

Figure 19: Additonal samples for the KTH dataset (cf. Figure 15). The subject in this example is boxing, which is the most challenging action in the dataset as all methods are far from the ground truth.

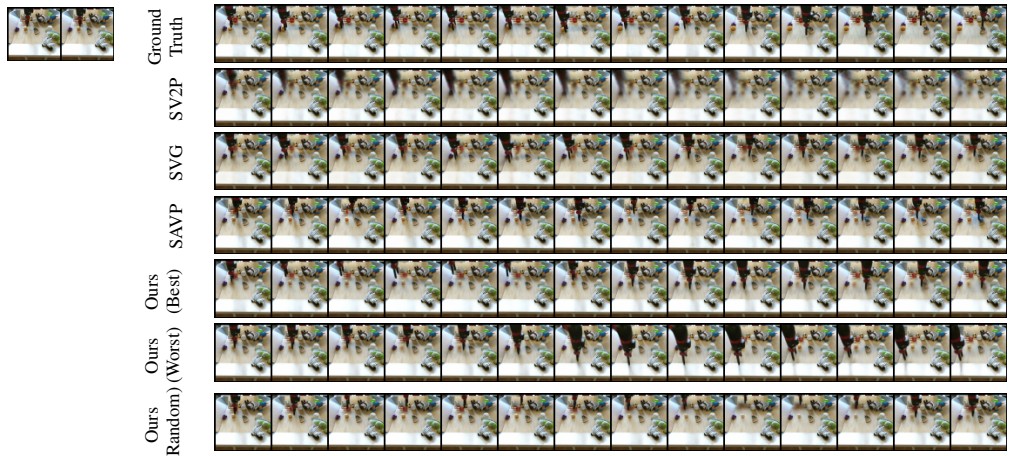

Figure 20: Conditioning frames and corresponding ground truth, best samples from SVG, SAVP and our method, and worst and random samples from our method, for an example of the BAIR dataset. Samples are chosen according to their LPIPS with respect to the ground truth.

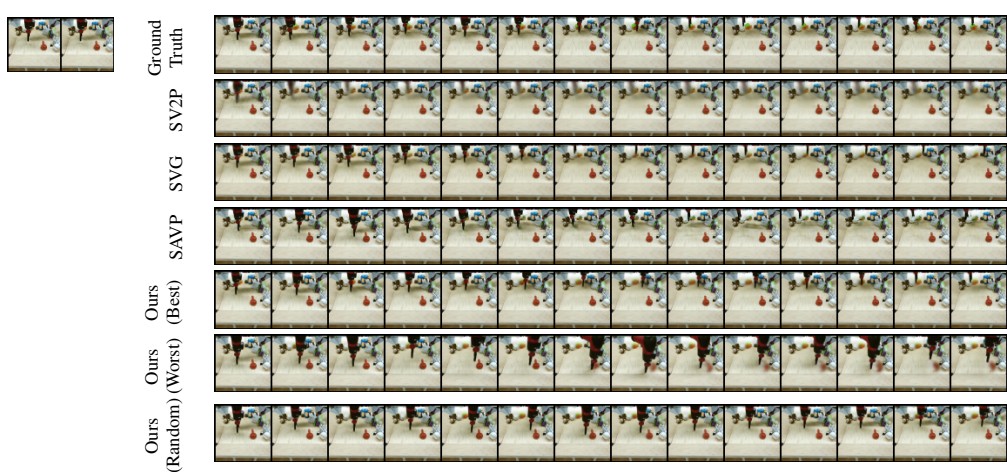

Figure 21: Additonal samples for the KTH dataset (cf. Figure 20).

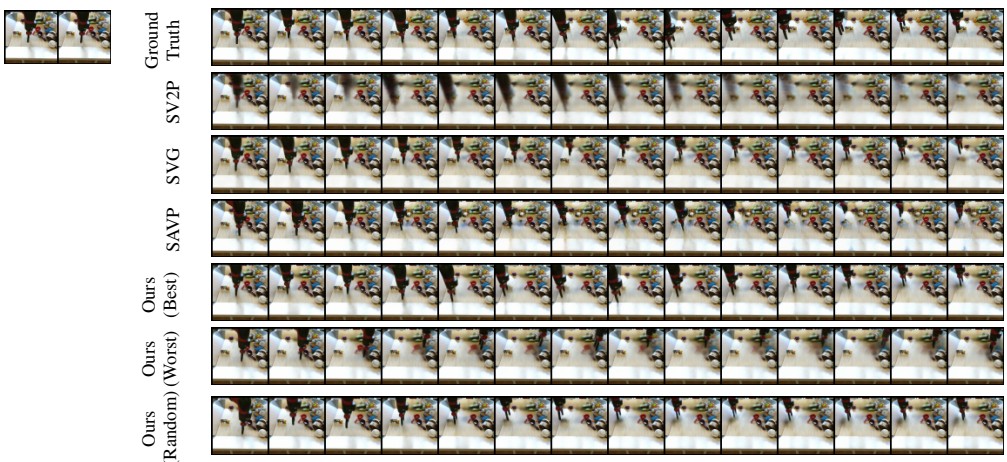

Figure 22: Additonal samples for the KTH dataset (cf. Figure 20).

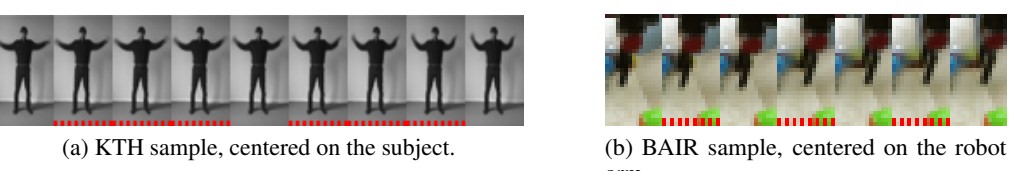

(a) KTH sample, centered on the subject.

(b) BAIR sample, centered on the robot arm.

Figure 23: Generation examples at doubled frame rate, using a halved $\Delta t$ compared to training. Frames including a bottom red dashed bar are intermediate frames.

## H.5 CONTENT SWAP

We present in Figures 24 to 28 additional examples of content swap as in Figure 8.

## H.6 INTERPOLATION IN THE LATENT SPACE

We present in Figures 29 and 30 additional examples of interpolation in the latent space between two trajectories.

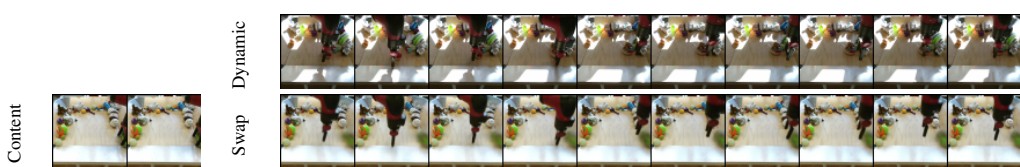

Figure 24: Video (bottom right) generated from the combination of dynamic variables $(\boldsymbol{y}, \boldsymbol{z})$ inferred with a video (top) and the content variable $(\boldsymbol{w})$ computed with the conditioning frames of another video (bottom left).

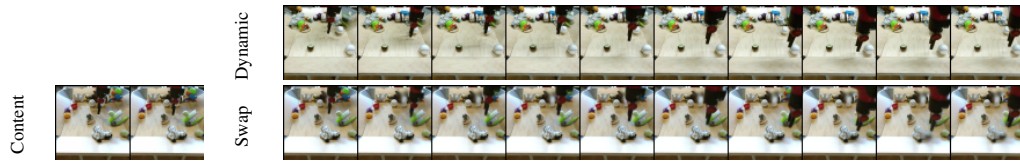

Figure 25: Additional example of content swap (cf. Figure 24).

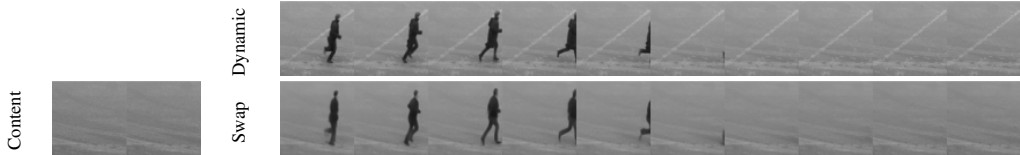

Figure 26: Additional example of content swap (cf. Figure 24). In this example, the extracted content is the video background, which is successfully transferred to the target video.

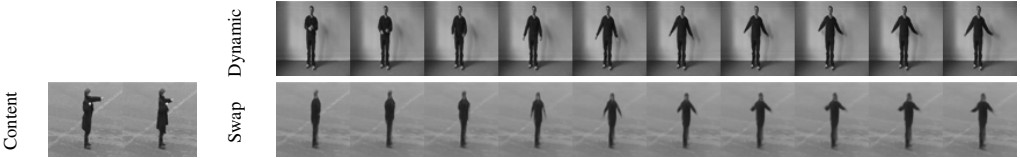

Figure 27: Additional example of content swap (cf. Figure 24). In this example, the extracted content is the video background and the subject appearance, which are successfully transferred to the target video.

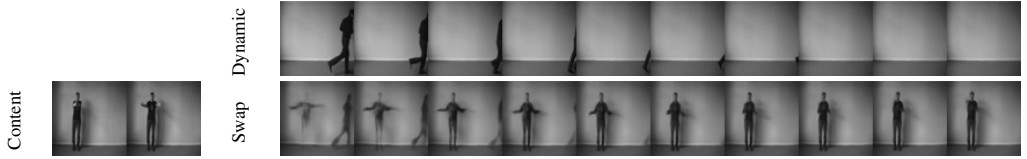

Figure 28: Additional example of content swap (cf. Figure 24). This example shows a failure case of content swapping.

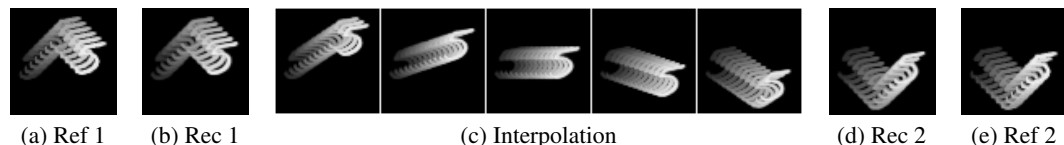

(a) Ref 1    (b) Rec 1                  (c) Interpolation                  (d) Rec 2    (e) Ref 2

Figure 29: From left to right, $x^s$, $\widehat{x}^s$ (reconstruction of $x^s$ by the VAE of our model), results of the interpolation in the latent space between $x^s$ and $x^t$, $\widehat{x}^t$ and $x^t$. Each trajectory is materialized in shades of grey in the frames.

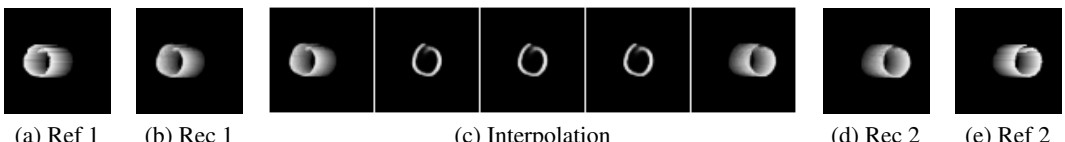

(a) Ref 1    (b) Rec 1                  (c) Interpolation                  (d) Rec 2    (e) Ref 2

Figure 30: Additional example of interpolation in the latent space between two trajectories (cf. Figure 29).

