# OpenReview forum: "Stochastic Latent Residual Video Prediction"
_ICLR.cc/2020/Conference — Reject_

### Official Review · AnonReviewer3 · 2019-10-22
**Official Blind Review #3**

**Rating:** 6

**Review:**

Contributions: this submission proposes a video pixel generation framework with the goal to decouple visual appearance and dynamics. The latent dynamics are modeled with a latent residual dynamics model. Empirical evaluations on moving MNIST show that the proposed residual dynamics model outperform MLP or GRU. On more challenging KTH and BAIR datasets, the proposed method achieves on par or better quantitative performance with previous methods, and have nice qualitative results on content "swap" and dynamics interpolation.

Assessment:
- To my knowledge, the proposed model is novel for video generation.
- The proposed method is supported with strong quantitative results and qualitative analysis, ablation on Moving MNIST shows that the proposed latent residual dynamics model outperforms MLP and GRU baselines.
- The authors might be interested in related work on video generation with decoupled appearance and dynamics models, such as [1]. It would also be interesting to see evaluation on more challenging datasets, such as Human3.6M.
- Question: how does the proposed inference framework make sure to decouple appearance with dynamics? Can y_i not encode the appearance information?


[1] Minderer et al., Unsupervised Learning of Object Structure and Dynamics from Videos. NeurIPS 2019.

-----------------------------
Post rebuttal:
Thank you for your answers to my questions and the updated manuscript. My questions have been addressed and the additional results further confirm the performance of the proposed method. Therefore I recommend weak accept of the submission.

**Experience Assessment:**

I have published one or two papers in this area.

**Review Assessment: Checking Correctness Of Derivations And Theory:**

I assessed the sensibility of the derivations and theory.

**Review Assessment: Checking Correctness Of Experiments:**

I assessed the sensibility of the experiments.

**Review Assessment: Thoroughness In Paper Reading:**

I read the paper at least twice and used my best judgement in assessing the paper.

---

> ### Author Response · Authors · 2019-11-15
> **Answer to Reviewer 3**
>
> We would like to thank you for your supportive feedback. We updated the submission following your remarks and answer your comments below.
>
>     - Decoupling appearance and dynamic:
>
>       Our model manages to decouple appearance and dynamics in the following way.
>       Firstly, the dynamics is made independent from the previously generated frames, in a state-space manner, as described in the manuscript.
>       Secondly, the content variable w is designed to remove as much visual information from y as possible. This relies on the fact that the dynamic variables y and z are regularized by KL, while the content variable w is not regularized in the loss function. w is instead prevented from containing any temporal information as it is inferred during training with randomly sampled frames and a permutation-invariant network. w is otherwise free to contain any non temporal information. The dynamic variables y and z are encouraged, because of the KL penalty, to contain only necessary information, and therefore should only contain temporal information that could not be captured by w.
>       We clarify this point in Section 3.2 of the revised submission.
>
>
>     - Human3.6M:
>
>       We thank the reviewer for suggesting this relevant dataset. We would have been pleased to show results of our method on this dataset. Unfortunately, it is not publicly available and we were only granted access only a couple of days ago. Please note, however, that the KTH dataset that we consider is somewhat similar to Human3.6M, as both datasets consist in videos of actions performed by subjects in front of a camera.
>
>
>     - Related work:
>
>       We thank the reviewer for pointing out the recent missing reference [1] that we included in the manuscript as related work along with discussion regarding differences with our work.
>       It is orthogonal to our work as they improve the VRNN dynamic model by using  frame-wise key-point representations instead of raw frames, while we focus on improving the dynamic model itself.
>
>
> We hope that we were able to answer your questions.
>
>
> [1] Minderer et al. Unsupervised Learning of Object Structure and Dynamics from Videos. In NeurIPS 2019.

---

### Official Review · AnonReviewer2 · 2019-10-23
**Official Blind Review #2**

**Rating:** 6

**Review:**

The paper proposes a video prediction model which explicitly decouples frame synthesis and motion dynamics. This is a very subtle change (compared to the current models) that can result in higher quality predictions.

First of all, the paper is extremely well written. It provides clear motivations and goals, as well as an impressively comprehensive related work that discusses their shortcomings. The experiments are comprehensive and provide good support for the claims. And finally, the appendix presents additional visualization and information.

On the main proposed method, it is a very subtle but reasonable change. Therefore, my suggestion to the authors is to provide a more thorough comparison with existing methods specifically SVG (Denton 2018) since the models share a lot of similarities. It is also quite similar to PlaNet (Hafner 2019). This is where the paper can be improved.

For the experiments, although they are quite comprehensive, there is still room for improvement. First, none of the metrics used are good evaluation metrics for frame prediction (I know they are quite common but that doesn't make them good) as they do not give us an objective evaluation in the sense of the semantic quality of predicted frames, specially for long videos. It really helps if the authors present additional quantitative evaluation to show that the predicted frames contain useful semantic information with metrics such as FVD and Inception score. Second, a pretribulation study is required to see where the improvements are coming from. Is it from a different architecture or the separation of dynamics?  Finally, a website with generated videos really helps for qualitative comparison!

Overall, this is a well-written paper with clear motivations and goals. I find the impact of the paper to be marginal (given the quality difference with already existing models) which can be improved by emphasizing more on other aspects such as disentanglement.

**Experience Assessment:**

I have published in this field for several years.

**Review Assessment: Checking Correctness Of Derivations And Theory:**

I carefully checked the derivations and theory.

**Review Assessment: Checking Correctness Of Experiments:**

I carefully checked the experiments.

**Review Assessment: Thoroughness In Paper Reading:**

I read the paper thoroughly.

---

> ### Author Response · Authors · 2019-11-15
> **Answer to Reviewer 2**
>
> We would like to thank the reviewer for his encouraging feedback and helpful suggestions. We updated the submission accordingly and provide answers and comments below.
>
>     - Impact and difference with SVG:
>
>       The difference between our model and SVG is twofold. Firstly, unlike SVG, our model decouples frame synthesis and dynamics, thanks to its state-space nature, a different prediction model and the use of a content variable. Secondly, our model introduces a stochastic residual update that is shown to significantly improve the performance of our model compared to a regular recurrent network.
>
>       Combined, these improvements allow our model to reach the state of the art of stochastic video prediction, besides obtaining desirable properties for its latent space that could not be achieved by prior models.
>       We think that these contributions could have substantial impact in the community, as the general principles of our model (state-space, residual dynamic, static content variable) can be generally applied to other models as well, even though this is out of the scope of this paper. For instance, replacing the VRNN model in [1] to model the evolution of key-points could bring additional performance gains. Moreover, the state-space nature of our model allows us to learn meaningful dynamic representations, typically used in model-based reinforcement learning [2], and illustrated in Figure 9.
>
>       We emphasized these points in the revised version of the paper in Sections 2, 4 and 5.
>
>
>     - Origin of improvements:
>
>       Our model shares the same encoder / decoder architecture as SVG, showing that its performance is due to our inference and dynamic systems rather than our choice of neural architecture.
>
>       To further study the origin of improvements, we presented, in the original submission, two baselines (GRU and MLP). The only difference between those baselines and our model is dynamic-related (they either use recurrent networks such as those of VRNN [3] or recurrent MLPs as an update method, instead of our residual updates). From our experiments, we reach two conclusions. First, all three versions of our method (residual, MLP, GRU) outperform prior methods. Therefore, this improvement is due to their common inference method, latent nature and content variable. Secondly, the residual update provide an additional gain of performance compared to the other methods, that can only be explained by the difference in the update method.
>
>       This analysis, that was originally performed on both deterministic and stochastic versions of Moving MNIST, have been confirmed by *new experiments on KTH*.
>
>       We clarified these points and included the new results in the revised version of the submission in Section 4 and Table 2.
>
>
>     - FVD metric:
>
>       We would like to thank the reviewer for suggesting this additional metric. We provide additional FVD scores for all tested methods on KTH and BAIR, as well as an analysis of these results in Table 2 and 3, and  Section 4. These results experimentally confirm our previous findings with respect to PSNR, SSIM, and LPIPS, i.e., we outperform the state of the art on KTH and are on par with state-of-the-art methods on BAIR. We believe that FVD is complementary to previously considered metrics, and we include an additional discussion on this matter in Section 4.
>
>
>     - PlaNet:
>
>       We thank the reviewer for pointing out the missing reference [4], that we added in the related work section. The differences with our model are the following. It is a control model that focuses and is evaluated on planning tasks which are out of scope of our paper. Also, [4] uses a dynamic model that is close to the one of [5] (with additional latent overshooting) based on recurrent neural networks, unlike our residual model.
>
>
>     - Website:
>
>       We provide an anonymized website at:  https://sites.google.com/view/srvp/. It hosts generated videos with qualitative comparisons, along with the corresponding anonymized code.
>
>
> We hope that we were able to answer the questions adequately and to alleviate raised concerns.
>
>
> [1] Minderer et al. Unsupervised Learning of Object Structure and Dynamics from Videos. In NeurIPS 2019.
> [2] Gregor et al. Temporal Difference Variational Auto-Encoder. In ICLR 2019.
> [3] Chung et al. A Recurrent Latent Variable Model for Sequential Data. In NIPS 2015.
> [4] Hafner et al. Learning Latent Dynamics for Planning from Pixels. In ICML 2019.
> [5] Fraccaro et al. Sequential Neural Models with Stochastic Layers. In NIPS 2016.

---

### Official Review · AnonReviewer1 · 2019-10-23
**Official Blind Review #1**

**Rating:** 3

**Review:**

Summary:

The paper proposes a video prediction method based on State-Space Models. The paper describes two main contributions:

1. By learning dynamics in the latent state space, the method avoids the high computational cost and accumulating image reconstruction errors of autoregressive models that condition on generated frames.

2. To model dynamics, the paper proposes a residual update rule inspired by Euler’s method to solve ODEs. According to this rule, the update to the latent state y_t is modeled as an additive residual f(y_t, z_{t+1}). This has the advantage that the step size of the discretization can be adjusted freely, e.g. between training and inference.

The paper provides extensive experimental comparison of their model to the SVG and SAVP models on several standard datasets. The paper further contains experiments illustrating features of the model such as disentangling dynamics and content, and interpolation of dynamics in the latent space.

Decision:

The paper is written clearly and the mathematical treatment and experiments appear rigorous. The idea of predicting video using state-space models is interesting and promising. However, as described below, the paper overstates its novelty and falls short of showing the advantages of the method beyond incremental improvements on frame-wise image quality metrics. I therefore suggest rejection in its current version.

Supporting arguments and suggestions:

1. The idea to use fully latent models for video prediction, to untie frame synthesis and dynamics, is not new and the paper does not fully cite this literature. For example, [1] and [2] perform unsupervised, non-autoregressive video prediction. The differences to these models should be discussed.

2. The advantages of the residual update rule are not made clear enough. The parallels to the ODE literature seem tenuous. The main advantage described in the paper is the ability to synthesize videos at different frame rates, but interpolation over such short time horizons is not a hard problem. At least, the paper should compare to existing methods for frame interpolation. Apart from interpolation (variable step size), it appears that the update rule could be changed from y_{t+1} = y_t + f(y_t, z_{t+1}) to y_{t+1} = f(y_t, z_{t+1}) without impact to the model. How is it different from the standard VRNN formulation [3]? More experiments to show the advantage of the proposed update rule would be helpful.

3. Some of the experiments seem like interesting starting points but do not support general claims. For example, Fig 2 (b) shows that the proposed dynamics model is better than an MLP or GRU on deterministic Moving MNIST, but is this also true on real datasets, which have much more complex dynamics? Similarly, the interpolation in Figure 9 is intriguing, but it would be helpful to describe and test how this ability is useful for applications of the predictive model.

4. The comparisons use frame-wise metrics of image quality (PSNR, SSIM, LPIPS). Even though they are common in the literature, these metrics are unsuitable for comparing long video sequences due to their stochasticity. The metrics are probably dominated by relatively uninteresting features such as the quality of the static background. Metrics for comparing entire videos exist (e.g. FVD [4]) and should be used. Even better, the paper should demonstrate the usefulness of the model for downstream tasks such as reinforcement learning, although I understand that this may be out of scope.

Minor comments:

- As far as I know, the correct term for error terms is residual, not residue.
- What do the error bars in the figures show? Please add this information to the figure legends.

[1] Wichers et al., 2018, https://arxiv.org/pdf/1806.04768.pdf
[2] Minderer et al., 2019, https://arxiv.org/abs/1906.07889
[3] Chung et al, 2015, https://arxiv.org/abs/1506.02216
[4] Unterthiner et al., 2018, https://arxiv.org/abs/1812.01717

**Experience Assessment:**

I have published one or two papers in this area.

**Review Assessment: Checking Correctness Of Derivations And Theory:**

I assessed the sensibility of the derivations and theory.

**Review Assessment: Checking Correctness Of Experiments:**

I carefully checked the experiments.

**Review Assessment: Thoroughness In Paper Reading:**

I read the paper at least twice and used my best judgement in assessing the paper.

---

> ### Author Response · Authors · 2019-11-15
> **Answer to Reviewer 1**
>
> We would like to thank the reviewer for his comprehensive and useful feedback. We updated the submission accordingly and provide answers and comments below.
>
>     - Missing references and discussions:
>
>       We thank the reviewer for pointing out missing references [1] and the recent [2]. We included them in the manuscript as related work along with discussions regarding differences with our work.
>
>       [1] tackles video prediction in a deterministic setting, while we focus on stochastic prediction instead. As its architecture doesn’t include stochastic components, it cannot produce a diversity of predictions by design.
>
>       [2] is orthogonal to our work as they improve the VRNN dynamic model by using  frame-wise key-point representations instead of raw frames, while we focus on improving the dynamic model itself. Specifically, we pointed out that re-using predicted frames to perform future frame predictions leads to prediction error accumulation over time and should be avoided. As [2] uses VRNN as a backbone dynamic model, it is autoregressive, albeit in the key-point space instead of the pixel space. While this change could mitigate the above-mentioned problem, the extent of such mitigation is unclear. We believe both approaches to be complementary. It would be an interesting future work to study the behavior of our model in the setting of [2] by replacing their VRNN-based temporal component by ours. This is, however, outside the scope of this paper.
>
>
>     - Generation at arbitrary frame rates:
>
>       We would like to bring to the reviewer’s attention that our model was not meant to serve as an interpolation method and is not trained as such. Instead, we presented experiments regarding the ability to generate at higher frame rates in order to analyze the dynamics learned by our model. The fact that our model maintains the same performance when generating at a higher framerate, without further training, shows that it learned a continuous dynamic driven by a piecewise ODE (i.e., the learned dynamic of each interval between two consecutive frames is driven by an ODE). We emphasized this point in Sections 3.1 and 4 of the manuscript revision.
>
>
>     - Further study of the update rule:
>
>       In addition to the previous experiments on deterministic and stochastic Moving MNIST, we performed further experiments on the KTH dataset to complement our analysis of the effects of the update rule. Results are presented in Table 2 and further discussed in Section 4 of the revised manuscript. They confirm the structural advantage in terms of performance of residual updates compared to other considered update methods (GRU and MLP). The MLP version better captures dynamics than GRU (in terms of PSNR and SSIM), but drops in terms of realism (captured by LPIPS and FVD). On the other hand, the residual version slightly improves this gain in terms of dynamics but significantly pushes further the realism of the generated videos. This observed advantage comes from the well documented structural prior that the residual update establishes in the network [3, 4].
>
>
>     - Interpolation of dynamics:
>
>       The experiment showing interpolation of dynamics in Figure 9 is designed to assess the interpretability and representation quality of the learned latent space. Such experiments are standard in the literature of generative models [5],  but could not have been proposed in the stochastic video prediction community before, as it takes advantage of the state-space nature of our model. We believe that these considerations are another interesting argument in favor of state-space generative models.
>
>
>     - FVD metric:
>
>       We would like to thank the reviewer for suggesting this additional metric. We provide additional FVD scores for all tested methods on KTH and BAIR, as well as an analysis of these results in Table 2 and 3, and  Section 4. These results experimentally confirm our previous findings with respect to PSNR, SSIM, and LPIPS, i.e., we outperform the state of the art on KTH and are on par with state-of-the-art methods on BAIR. We believe that FVD is complementary to previously considered metrics, and we include an additional discussion on this matter in Section 4.
>
>
>     - We changed the wording from “residue” to “residual”.
>
>     - The error bars correspond to the 95%-confidence interval of the plotted mean, similarly to [6]. We added this information in the figure captions.
>
>
> We hope that we were able to alleviate raised concerns.
>
>
> [1] Wichers et al. Hierarchical Long-term Video Prediction without Supervision. In ICML 2018.
> [2] Minderer et al. Unsupervised Learning of Object Structure and Dynamics from Videos. In NeurIPS 2019.
> [3] Chen et al. Neural Ordinary Differential Equations. In NeurIPS 2018.
> [4] He et al. Deep Residual Learning for Image Recognition. In CVPR 2016.
> [5] Kingma et al. Glow: Generative Flow with Invertible 1x1 Convolutions. In NeurIPS 2018.

---

### Author Response · Authors · 2019-11-15
**Rebuttal and Revision**

We would like to thank the reviewers for their comprehensive, insightful and encouraging comments. We answered individually to each of them. Below, we summarize the main changes made in the revision of the manuscript.

    - We added FVD scores for all compared methods in Tables 2 and 3, as well as an additional analysis in Section 4. We outperform other methods on this metric on the KTH dataset, and are on a par with the state of the art on BAIR.

    - We extend the study of our dynamic model to the KTH dataset on Section 4, and reported new results on Table 2. They confirm our findings on the Moving MNIST dataset showing the structural advantage of residual dynamics.

    - We emphasized the role of the content variable in decoupling dynamic model and visual features in Section 3.2 with an additional explanation.

    - We further clarified the differences between our method and SVG in Sections 2 and 4.

    - We added references [1, 2, 3] to the related work with additional discussions.

We would like to highlight the anonymized project webpage containing the code corresponding to our method as well as video samples: https://sites.google.com/view/srvp/.

[1] Minderer et al. Unsupervised Learning of Object Structure and Dynamics from Videos. In NeurIPS 2019.
[2] Wichers et al. Hierarchical Long-term Video Prediction without Supervision. In ICML 2018.
[3] Hafner et al. Learning Latent Dynamics for Planning from Pixels. In ICML 2019.

---

### Author Response · Authors · 2020-03-05
**New Version & Changes**

A new version of our article was accepted to ICML 2020 (https://proceedings.icml.cc/book/3249.pdf), with the following main improvements:
    - we included new results on the challenging Human3.6M dataset, showing that we outperform the state-of-the-art StructVRNN [1];
    - we computed FVD scores for all tested models;
    - we clarified the experimental settings and interpretation of the experiment regarding generation at different frame rates;
    - we improved our results and updated their analysis for the BAIR dataset;
    - we publicly released the fully documented code at https://github.com/edouardelasalles/srvp;
    - we publicly released the pretrained models  at https://data.lip6.fr/srvp/.

[1] Minderer et al. Unsupervised Learning of Object Structure and Dynamics from Videos. In NeurIPS 2019.

---

### Decision · Program_Chairs · 2019-12-19

**Decision:**

Reject

**Comment:**

The paper proposes a method for learning a latent dynamics model for videos. The main idea is to learn a latent representation and model the dynamics of the latent features via residual connection motivated by ODE. The architectural choice of residual connection itself is not new as many prior works have employed "skip connections" in hidden representations but the notion of connecting this with ODE and factoring time as input into the residual function seems a new idea. The experimental results show the promise of the proposed method on moving MNIST, KTH, and BAIR datasets. The experiments on different frame rates are also nice.  In terms of weakness, the evaluation is performed on relatively simple domains (e.g., moving MNIST and KTH) with static backgrounds and the improvement on BAIR dataset (which is not considered as a difficult benchmark) in terms of FVD is not as clear. For the BAIR dataset, it's unclear how the proposed method will handle the interactions between the robot arm and background objects due to the modeling assumption (i.e., static background). In this sense, content swap results on BAIR dataset look quite anecdotal, and the significance is limited. For improvement, I would suggest adding evaluations on other challenging domains, such as Human 3.6M (where human motions are much more uncertain compared to KTH) and other Robot datasets with more complex robot-object interactions. Overall, the paper proposes an interesting architecture with promising results on relatively simple datasets, but the advantage over existing SOTA methods on challenging benchmarks is unclear yet.